# Uncertainty and sensitivity analysis for probabilistic weather and climate risk modelling: an implementation in CLIMADA v.3.1.0

Chahan M. Kropf[1,2], Alessio Ciullo[1,2], Laura Otth[1], Simona Meiler[1,2], Arun Rana[3], Emanuel Schmid[1], Jamie W. McCaughey[1,2], and David N. Bresch[1,2]

[1]Institute for Environmental Decisions, ETH Zurich, Universitätstr. 16, 8092 Zurich, Switzerland
[2]Federal Office of Meteorology and Climatology MeteoSwiss, Operation Center 1, P.O. Box 257, 8058 Zurich-Airport, Switzerland
[3]Frankfurt School of Finance and Management gemeinnützige GmbH, Adickesallee 32-34, 60322 Frankfurt am Main, Germany

**Correspondence:** Chahan M. Kropf (chahan.kropf@usys.ethz.ch)

**Abstract.** Modelling the risk of natural hazards for society, ecosystems, and the economy is subject to strong uncertainties, even more so in the context of a changing climate, evolving societies, growing economies, and declining ecosystems. Here we present a new feature of the climate risk modelling platform CLIMADA which allows to carry out global uncertainty and sensitivity analysis. CLIMADA underpins the Economics of Climate Adaptation (ECA) methodology which provides decision makers with a fact-base to understand the impact of weather and climate on their economies, communities, and ecosystems, including appraisal of bespoke adaptation options today and in future. We apply the new feature to an ECA analysis of risk from tropical cyclone storm surge to people in Vietnam to showcase the comprehensive treatment of uncertainty and sensitivity of the model outputs, such as the spatial distribution of risk exceedance probabilities or the benefits of different adaptation options. We argue that broader application of uncertainty and sensitivity analyses will enhance transparency and inter-comparison of studies among climate risk modellers and help focus future research. For decision-makers and other users of climate risk modelling, uncertainty and sensitivity analysis has the potential to lead to better-informed decisions on climate adaptation. Beyond provision of uncertainty quantification, the presented approach does contextualise risk assessment and options appraisal, and might be used to inform the development of story-lines and climate adaptation narratives.

## 1 Introduction

Societal impacts from natural disasters have steadily increased over the last decades (IFRC, 2020), and they are expected to follow the same path under climatic, socio-economic, and ecological changes in the coming decades (Masson-Delmotte et al., 2021). This creates the need for better preparedness and adaptation towards such events, and raises a demand for risk assessments and adaptation options appraisal studies at the local, national and global levels. Typically, such studies are carried out through the use of computer models – which will be referred to as climate risk models in this article – that allow to estimate

the socio-economic and ecological impact[1] of various natural hazards such as tropical cyclones, wildfires, heat waves, droughts, coastal, fluvial, or pluvial flooding.

The specific set-up of climate risk models depends on the hazard under consideration, the location of interest and the study's goal. Such models however often share a similar structure given by three sub-models usually referred to as *hazard*, *exposure* and *vulnerability*. These constitute the input variables of climate risk models and represent the main drivers of climate risk as defined by the Intergovernmental Panel on Climate Change (IPCC) (Pachauri et al., 2015). Hazard is a model of the physical forcing at each location of interest, exposure is a model of the spatial distribution of the exposed elements such as people, buildings, infrastructures and ecosystems, and vulnerability is characterized by a uni- or multi-variate impact function describing the impact of the considered hazard on the given exposed elements. By combining hazard, exposure and vulnerability, the socio-economic impact of natural hazards can be assessed. In so doing, one can also carry out an adaptation options appraisal by comparison of the current and future risk reduction capacity of adaptation options with expected implementation costs.

In practice, the quantification of risk with climate risk models are particularly challenging as they involve dealing with the absence of robust verification data (Matott et al., 2009; Pianosi et al., 2016; Wagener et al., 2022) when setting-up the hazard, exposure and vulnerability sub-models, as well as dealing with large uncertainties in the input parameters and the model structure itself (Knüsel, 2020). For example, in hazard modelling, many authors have shown large uncertainties affecting the computation of flood maps through hydraulic modelling (Merwade et al., 2008; Dottori et al., 2013) and, similarly, alternative models have been proposed for modelling tropical cyclones tracks and intensities (Emanuel, 2017; Bloemendaal et al., 2020). For exposure, notable uncertainties are associated with the quality of the data being used, their resolution and, as often proxy data are used (Ceola et al., 2014; Eberenz et al., 2020), their fitness-for-purpose. The vulnerability module also introduces significant uncertainties, because data needed to calibrate impact function curves are often very scarce and scattered (Wagenaar et al., 2016). In addition, uncertainties affecting exposure, hazard and vulnerability are exacerbated by the unknowns in climatic, economical, social and ecological projections. Furthermore, modelling adaptation options is a process that is particularly strongly affected by normative uncertainties (Knüsel et al., 2020). For example, the choice of the discount rate, which does affect the effectiveness of a given option, raises inter-generational justice issues (Doorn, 2015; Moeller, 2016; Mayer et al., 2017). Finally, the choice of output metrics, the performance measures and the very formulation of the risk management problem also underlie value-laden choices (Kasprzyk et al., 2013; Ciullo et al., 2020), as they dictate what actors and what actors' interests are included in the risk assessment and adaptation options appraisal (Knüsel et al., 2020; Otth, 2021; Otth et al., 2022).

Among the established methods proposed by the scientific literature to quantitatively treat uncertainties in model simulation are uncertainty and sensitivity analysis (Saltelli, 2008). While for both methods an analytical treatment is preferable (Norton, 2015), it is often not possible. Therefore, numerical Monte-Carlo or Quasi-Monte-Carlo schemes (Lemieux, 2009; Leobacher and Pillichshammer, 2014) are applied, which require repeated model runs using different values for the uncertain input pa-

---

[1]"Impacts generally refer to effects on lives; livelihoods; health and well-being; ecosystems and species; economic, social and cultural assets; services (including ecosystem services); and infrastructure. Impacts may be referred to as consequences or outcomes, and can be adverse or beneficial." (Field et al., 2014)

rameters. Uncertainty analysis is then the study of the distribution of outputs obtained when the uncertain input parameters are sampled from plausible uncertainty ranges. Ideally, these plausible ranges should be defined based on background knowledge related to these parameters (Beven et al., 2018b). Sensitivity analysis in turn assesses the respective contributions of the input parameters to the total output variability, and often builds upon uncertainty analysis. It allows to test the robustness of the model, to single out the input uncertainties most responsible for the output uncertainty, and to improve understanding about the model's structure and input-output relationships (Pianosi et al., 2016). Arguably, conducting uncertainty and sensitivity analysis should be part of any modeling exercise as it reveals its fitness for purpose and limitations (Saltelli et al., 2019). Nevertheless, an uncertainty and sensitivity analysis is still lacking in many published modelling studies (Beven et al., 2018a; Saltelli et al., 2019). In this context, climate risk assessment studies are no exception. Although there are examples in the scientific literature of applications of uncertainty and sensitivity analysis to the full (de Moel et al., 2012; Koks et al., 2015) or partial (Hall et al., 2005; Savage et al., 2016) climate risk modeling chains, these techniques (Douglas-Smith et al., 2020) are neither common practice, nor applied in a systematic fashion. This may strongly undermine the quality of the risk assessment and adaptation options appraisal, and may lead to poor decisions (Beven et al., 2018a).

In order to fill this gap and facilitate the widespread adoption and application of uncertainty and sensitivity analysis in climate risk models, this article introduces and showcases a new feature of the probabilistic climate risk assessment and modelling platform CLIMADA (CLIMate ADAadaptation) (Aznar-Siguan and Bresch, 2019; Bresch and Aznar-Siguan, 2021; Kropf et al., 2022) which seamlessly integrates the *SALib - Sensitivity Analysis Library in Python* package (Herman and Usher, 2017) into the overall CLIMADA modeling framework, and thus supports all sampling and sensitivity index algorithms implemented therein. The new feature allows conducting uncertainty and sensitivity analysis for any CLIMADA climate risk assessment and adaptation options appraisal with little additional effort, and in a user-friendly manner. Here we describe the UNcertainty and SEnsitity QUAntification (unsequa) module in detail and demonstrate it's use on a previously published case study on the impact of tropical cyclones in Vietnam (Rana et al., 2021).

The article is structured as follows: Sect. 2 will introduce the CLIMADA modeling platform and describe how uncertainty and sensitivity analysis are integrated therein; Sect. 3 demonstrates the use of uncertainty and sensitivity analysis by revisiting a case study on tropical cyclone impact in Vietnam ; Sect. 4 discusses results and provides an outlook.

## 2 Uncertainty and sensitivity analysis in the climate risk modeling platform CLIMADA

### 2.1 Brief introduction to CLIMADA

To our knowledge, CLIMADA is the first global platform for probabilistic multi-hazard risk modelling and options appraisal to seamlessly include uncertainty and sensitivity analysis in its workflow as described in this section. CLIMADA is written in *Python 3* (Van Rossum and Drake, 2009), and is fully open-source and open-access (Kropf et al., 2022). It implements a probabilistic multi-hazard global natural catastrophe impact model based on the three sub-modules hazard, exposure, and vulnerability . It can be used to assess the risk of natural hazards and to perform adaptation options appraisal by comparison

of the averted impact (benefit) thanks to adaptation measures of any kind (from grey to green infrastructure, behavioral, etc.) with their implementation costs (Aznar-Siguan and Bresch, 2019; Bresch and Aznar-Siguan, 2021).

The hazard is modelled as a probabilistic set of events, each one a map of intensity at geographical locations, and with an associated probability of occurrence. For example, the intensity can be expressed in terms of flood depth in meters, maximum wind speed in meters per second, or heat wave duration in days, and the probability as a frequency per year. The exposure is modelled as values distributed on a geographical grid. For instance, the number of animal species, the value of assets in dollars, or the number of people living in a given area. The vulnerability is modelled for each exposure type by an impact function , which is a function of hazard intensity (for details, see Aznar-Siguan and Bresch, 2019). This could be e.g., a sigmoid function with $0\%$ of affected people below 0.2m flood depth, and $90\%$ of affected people above 1m flood depth. The adaptation measures are modelled as modification of the impact function, exposure or hazard. For example, a new regional plan can incite people to relocate to less flood-prone areas, hence resulting in a modified exposure (c.f. Aznar-Siguan and Bresch, 2019; Bresch and Aznar-Siguan, 2021).

The risk of a single event is defined as its impact multiplied by its probability of occurrence. The impact is obtained by multiplying the value of the impact function at a given hazard intensity with the exposure value at a given location. The total risk over time is obtained from the impact matrix , which entails the impact of each hazard event at each exposure location, and the hazard frequency vector. The benefits of adaptation measures is obtained as the change in total risk. Both the total risk and the benefits can thus be computed for today and in the future, following climate change scenarios and socio-economic development pathways (c.f. Aznar-Siguan and Bresch, 2019; Bresch and Aznar-Siguan, 2021).

With CLIMADA, risk is assessed in a globally consistent fashion; from city to continental scales; for historical data or future projections; considering various adaptation options ; including future projections for the climate, socio-economic growth or vulnerability changes.

## 2.2 UNcertainty and SEnsitivity QUAntification (unsequa) module overview

The general workflow of the new uncertainty and sensitivity quantification module unsequa, illustrated in Fig. 1, follows a Monte-Carlo logic (Hammersley, 1960) and implements similar steps as generic uncertainty and sensitivity analysis schemes (Pianosi et al., 2016; Saltelli et al., 2019). It consists of the following steps:

– *Input variables and input parameters definition*. The probability distributions of the uncertain input parameters (random variables) are defined. They characterize the input variables – hazard, exposure and impact function for risk assessment, and additionally adaptation measure for adaptation options appraisal – of the climate risk model CLIMADA.

– *Samples generation*. Samples of the input parameter values are drawn according to their respective uncertainty probability distribution.

– *Model output computation*. The CLIMADA engine is used to compute all relevant model outputs for each of the samples for risk assessment (risk metrics) and/or adaptation options appraisal (benefit and cost metrics).

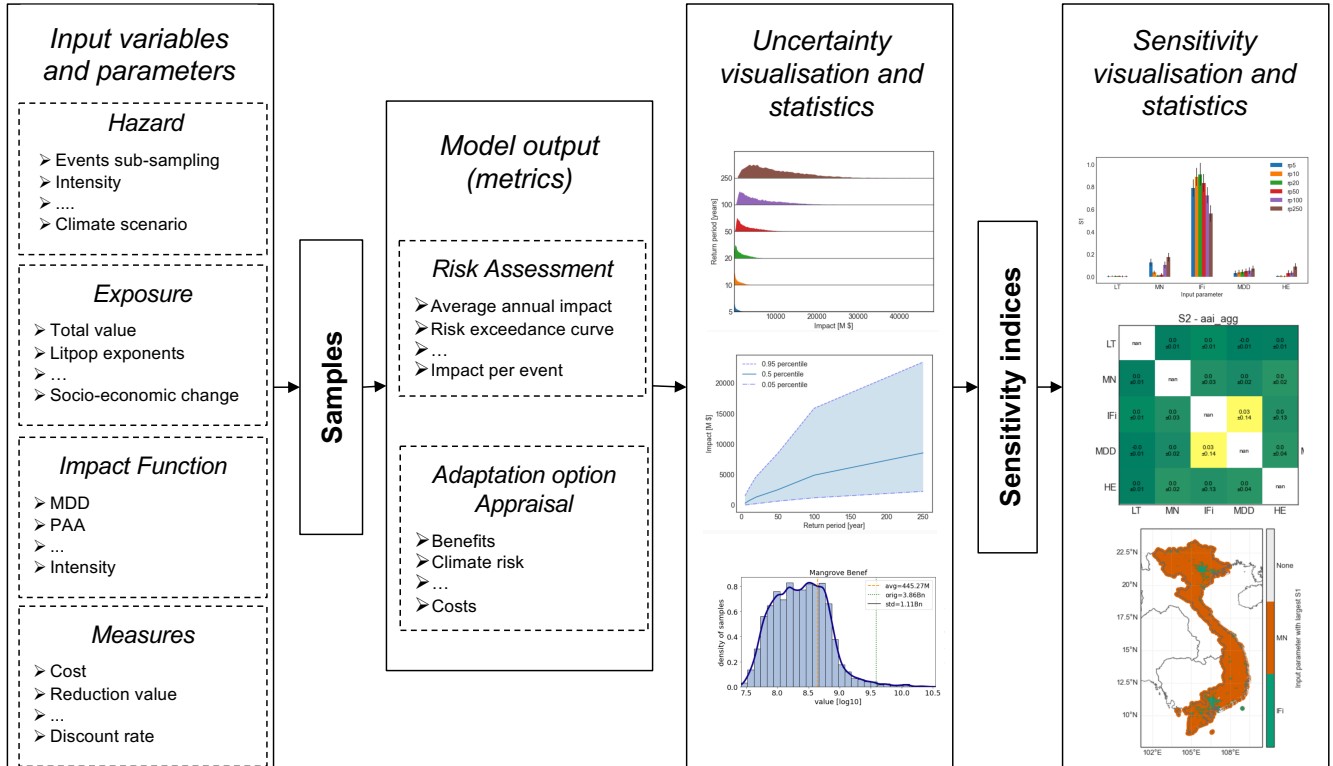

**Figure 1.** The workflow for uncertainty and sensitivity analysis with the unsequa module in CLIMADA consists of six steps (from left to right). 1. Define the input variables (hazard, exposure, impact function, adaptation measure) and their uncertainty input parameters (e.g., hazard intensity, total exposure value, impact function intensity, measures cost). 2. Generate the input parameter samples. 3. Compute the model output metrics of interest for risk assessment and adaptation options appraisal for each sample using the CLIMADA engine. 4. Analyse the obtained uncertainty distributions with statistical tools and provide a set of visualisations. 5. Compute the sensitivity indices for each input parameter and each output metric. 6. Analyse the sensitivity indices by means of statistical methods and provide different visualisations.

– *Uncertainty visualisation and statistics.* The distribution of model outputs obtained in the previous step are analyzed and visualized.

– *Sensitivity indices computation.* Sensitivity indices for each input parameter are computed for each of the model output metric distributions.

– *Sensitivity visualisation and statistics.* The various sensitivity indices are analyzed and visualized.

We remark that typically the third and fourth steps constitute the core elements of the uncertainty analysis, and the fifth and sixth steps the core elements of the sensitivity Analysis. In Sect. 2.3 we describe each one of the steps in more detail. A detailed documentation on how to use the unsequa module is available at https://climada-python.readthedocs.io/.

## 2.3 unsequa module detailed workflow

### 2.3.1 Input Variables and Parameters

The CLIMADA engine integrates the input variables exposure ($E$), hazard ($H$), and impact function ($F$) for risk assessment. For the adaptation options appraisal, the exposure and impact function are combined with the adaptation measure ($M$) in a container input variable called entity ($T$). Note that further input variables might be added in future versions of CLIMADA. Each of these input variables comes with any number of uncertainty input parameters $\alpha$, distributed according to an independent probability distribution $p_\alpha$. An input variable can have any number of uncertainty input parameters, and there is no restriction on the type of probability distributions (e.g. uniform, Gaussian, skewed, heavy-tailed, discrete, etc.). In the current implementation any distribution from the *Scipy.stats* Python module (Virtanen et al., 2020) is accepted. The input parameters can define any variation or perturbation of the input variables (e.g., initial conditions, boundary conditions, forcing inputs, resolutions, normative choices, etc.). [2] Note that the choice of the variation and the associated range and distribution can substantially affect the results of an uncertainty and sensitivity analysis (Paleari and Confalonieri, 2016). Ideally this modelling choice should be made based on solid background knowledge. However, the latter is often lacking or highly uncertain; in such cases, we encourage users to explore how the results may vary with alternate distributions and choices of input parameters. It is thus not always about deriving definitive quantitative values describing the deviation of the climate risk model's output from the "real" value, but also about assessing the robustness, sensitivity and plausibility of the model output under clearly defined assumptions.

Overall, the user must define one method for each of the uncertain input variables $X$, which returns the input variable's value $X(\alpha_1, \alpha_2, \ldots)$ for each valid value of the associated uncertain input parameters $\alpha_1, \alpha_2, \ldots$. The latter are univariate random variables distributed according to $p_{\alpha_1}, p_{\alpha_2}, \ldots$. In order to support the user, a series of helper method are implemented in the unsequa module (c.f. Appendix B). This general problem formulation allows for generic uncertainty parametrization, with broadly speaking two types of approaches: an input variable is directly perturbed with statistical methods, or the underlying model used to generate the input variable is fed with the uncertain parameters. Note that each input variable is independent, and thus either approach can be used for different input variables for the same study (c.f. the illustration case study in Section 3.1)

As one example, suppose we are modelling the impact of heat waves on people in Switzerland. As exposure layer we might use gridded population data based on the total population estimate from the UN World population prospect (Nations, 2019) reported to be $t_s = 8,655,000$ in 2020. Assuming an estimation error of $\pm 5\%$, the input variable $E$ has one uncertain input parameter $t$ with a uniform distribution $p_t$ between $[0.95 t_s, 1.05 t_s]$. As hazard we might consider the heat waves of the past 40 years as measured by the Swiss Meteorological institute. Disregarding measurement uncertainties, one could decide to model this without uncertainty. Finally, the impact function might be represented by a sigmoid function calibrated on past events which yields uncertainty for the slope $s$ and the asymptotic value $a$. The slope's uncertainty could be a multiplicative factor

---

[2]In literature, the terminology "input factor" instead of "input parameter" is also used. Here we shall use exclusively the terminology "input parameter" for numerical random variables, and "input variable" for the inputs to the CLIMADA model.

$s$ drawn from a truncated Gaussian distribution $p_s$ with mean 1, standard deviation 0.2, and the truncation of negative values, while the asymptotic value could be given by $a$ which follows a uniform distribution $p_a$ between $[0.8, 1]$.

As another example, we are interested in the risk of floodplain flooding for gridded physical assets in the Congo basin. The flood hazard is generated from a floodplain modeling information system (FMIS) with uncertainty parameters describing the uncertainty in the geospatial data, the temporal data, the model parameters (Mannings) and the hydraulic structure, such as shown in Merwade et al. (2008). These input parameters are used directly as uncertainty input parameters for the unsequa model, with a wrapper method returning a CLIMADA hazard object produced from the FMIS flood inundation map. In addition, the exposures are obtained by interpolating and down-scaling satellite images to a resolution $r$ (Eberenz et al., 2020). The sensitivity and robustness to the resolution choice is modelled by pre-computing exposures at resolutions $r = 50as, 100as, 150as, 300as, 1000as$. The uncertainty parameter is then $r$, with a uniform choice distribution between the pre-computed values. Finally, we consider all assets to be described by a single impact function, which is derived from three different case studies found in literature. The impact function's uncertainty is defined as a uniform choice distributed parameter $u \in [1, 2, 3]$ corresponding to the selection of one of the three impact functions.

Defining the appropriate input variable uncertainty definition, and identifying the relevant input parameters for a given case study is not a trivial task. In general, only a small subset of all possible parameters can be investigated Dottori et al. (2013); Pianosi et al. (2016). In order to identify the relevant parameters and defining the input variables' uncertainty accordingly, one can for instance use an assumption map (Knüsel et al., 2020), as presented for CLIMADA in Otth (2021); Otth et al. (2022). Another general strategy is to proceed iteratively: a first broad sensitivity analysis is used to identify the most likely important uncertainties, followed by a more detailed uncertainty and sensitivity analysis for full quantification.

### 2.3.2 Samples

In general, there are two basic approaches regarding how samples can be drawn. In the local 'one at a time' approach, the input parameters are varied one after another, keeping all the others constant (Pianosi et al., 2016). Local methods are conceptually simpler, but do neither capture interactions between input parameters nor non-linearities (Douglas-Smith et al., 2020). By contrast, in global methods, the input parameters are sampled from the full space at once (Matott et al., 2009). This allows for a more comprehensive depiction of model uncertainty by accounting for the interactions among the input parameters. Saltelli et al. (2019) even argue that uncertainty and sensitivity analysis should always be based on global methods for models with non-linearities such as CLIMADA.

Hence, the basic premise of the unsequa module is to use a global sampling algorithm based on (quasi-) Monte-Carlo sequences (Lemieux, 2009; Leobacher and Pillichshammer, 2014) to generate a set of $N$ samples of the input parameters. Here one sample refers to one value for each of the input parameters. Following the heat wave example described in Sect. 2.3.1, one would create $N$ global samples $x_n = (t_n, s_n, a_n)$ with $n \in [1, \ldots, N]$. One sample thus corresponds to a set of three numbers in this case. Choosing the correct number of samples is a notoriously difficult task (Iooss and Lemaître, 2015; Sarrazin et al., 2016). One generic approach is to start with a sample size that one can afford to generate reasonably efficiently (e.g., $N \sim 100D$), and then check the confidence intervals of the estimated sensitivity indices (c.f. Section 2.3.5). If relative values of

the estimated indices are too ambiguous due to the overlap of confidence intervals to draw key conclusions, one should either generate more samples, or use a more frugal method (e.g., reduce the number of input parameters) (Sarrazin et al., 2016).

CLIMADA imports the (quasi-) Monte-Carlo sampling algorithms from the *SALib* Python package (Herman and Usher, 2017). Thus all sampling algorithms from this package are directly available to the user within the unsequa module. These algorithms are all at least implemented for a uniform distribution over $[0, 1]$. In order to accommodate for any input parameter distributions, the unsequa module uses the percent point function of the target probability density distribution (c.f. Appendix A).

### 2.3.3   Model output: Risk assessment and Adaptation options appraisal

For each sample of the input parameters, the model output metrics are computed using the CLIMADA engine, e.g. for the risk assessment the impact matrix $I_n$ for each sample $x_n$. Following the heat wave example from the previous section, for each sample $x_n = (t_n, s_n, a_n)$ of the input parameters, the algorithm first sets the input variables $E(n) = E(t_n), H(n) = H$ and $I(n) = I(s_n, a_n)$. Second, the corresponding impact matrix $I_n$ is computed for each sample independently following the algorithm described in Aznar-Siguan and Bresch (2019). All CLIMADA risk output metrics such as the average annual impact, the exceedance frequency curve or the largest event are then derived from the matrix $I_n$ and the hazard frequency defined in $H(n)$.

Similarly, for the adaptation options appraisal, each sample is assigned with the corresponding input variables. Then the CLIMADA engine is used to compute the impact matrix $I_n^m(y)$ for each sample $n$, each adaptation measure $m$ and each year $y$ following the algorithm described in Bresch and Aznar-Siguan (2021). All CLIMADA benefit and cost metrics such as the total future risk, the adaptation measures benefits, the risk transfer options, and costs are derived from the impact matrix $I_n^m(y)$, the adaptation measure $M_m(n, y)$, the exposure $E(n, y)$ and the hazard $H(n, y)$. Note that in practice the input variables for the exposure, impact function, and adaptation measure are combined into one input variable called Entity $T(n, y)$, which also includes information about optional discount rates and risk transfer options.

We remark that no direct evaluation of the convergence of this quasi Monte-Carlo scheme is provided in the unsequa module, as it is not generally available for all the possible sampling algorithms available through the *SALib* package. Instead, the sensitivity analysis algorithms, to be described in Sect. 2.3.5 below, provide confidence intervals. In SALib, confidence intervals relate to the bounds which cover 95% of the possible sensitivity index value, estimated through bootstrap resampling. These can be used as a proxy to assess the convergence of the uncertainty analysis. If the intervals are large and overlapping, the result is likely not robust and the number of samples should be increased.

In all of the uncertainty and sensitivity analysis, computing the model outputs is usually the computationally most expensive step. For convenience, an estimation of the total computation time for a given run is thus provided in the unsequa module. Experiments showed that the computation time scales approximately linearly with the number of samples $N$, and is proportional to the time for a single impact computation. The latter is mostly defined by the size of the exposure (i.e., depends on the resolution, size of the considered geographical area, ...) and the size of the hazard (i.e., depends on the number of events, the centroid's resolution, etc.). In case the input variables are generated using an external model (e.g., a hydrological flood model

for the hazard), the computation time is also proportional to the external model run time. For complex models, this can be pro-
hibitively long. In such case, one can pre-compute the samples for the given input variable, thus trading CPU time for memory
(c.f. Litpop example in Section 3.2.1, and the helper methods in Appendix B). The number of samples $N$ in turn scales with
the dimension $D$ (i.e., the number of input parameters) depending on the chosen sampling method. For the default unsequa
module Sobol′ method, the scaling is $O(D)$. In addition, for the adaption options appraisal, the risk computation is repeated
for each of the $N_m$ adaptation measures. This results in a total computation time scaling of $O(D)$ for the risk assessment,
and of $O(D \cdot N_m)$ for the adaptation options appraisal. Thus, for large number of input parameters, and/or long single impact
computation times, and/or large numbers of adaptation measures, the computation time might become intractable. In this case,
one could consider using surrogate models (Sudret, 2008; Marelli and Sudret, 2014), a feature that might be added to future
iterations of the unsequa module.

### 2.3.4 Uncertainty visualisation and statistics

The output metrics values for each sample are characterized and visualized. To this effect, various plotting methods have been
implemented as shown in Sect. 3.2.5 and 3.3.5. It is, for instance, possible to visualize the full distributions, or compute any
statistical value for each model output metric. The key objective is to obtain an understanding of the uncertainties in the model
outputs beyond the mean value and standard deviation.

### 2.3.5 Sensitivity indices

The sensitivity index $S_\alpha(o)$ is a number that subsumes the sensitivity of a model output metric $o$ to the uncertainty of input
parameter $\alpha$ (Pianosi et al., 2016). Since CLIMADA is a non-linear model, only global sensitivity indices are suitable (Saltelli
and Annoni, 2010). To derive such global sensitivity indices, several algorithms are made available through the *SALib* Python
package (Herman and Usher, 2017), including variance-based (ANOVA) (Sobol′, 2001), elementary effects (Morris, 1991),
derivative based (Sobol' and Kucherenko, 2009), FAST (Cukier et al., 1973), and more (Saltelli, 2008). Importantly, each
method requires a specific sampling sequence to compute the model output distribution and results in distinct sensitivity indices.
These distinct indices typically will agree on the general findings (e.g., what input parameter has the largest sensitivity), but
might differ in the details as they correspond to fundamentally different quantities (e.g. derivatives against variances). The
recommended pairing of sampling sequence and sensitivity index method is described in the *SALib* documentation, and simple
save-guard checks have been implemented in the unsequa module. Note that it is technically valid to use different sampling
algorithms for the uncertainty, and for the sensitivity analysis. For example, one can first use sampling algorithm A to perform
an uncertainty analysis, i.e., steps from Sects. 2.3.1 - 2.3.4. Then, use another sampling algorithm B as required for the chosen
sensitivity index algorithm to perform the sensitivity analysis, i.e., steps from Sects. 2.3.1-2.3.3 and 2.3.5, 2.3.6. However, in
practice, since generating samples is often the computational-time bottleneck, it is more convenient to use the same methods
so that the same samples can used for both steps (Borgonovo et al., 2017).

For typical case-studies using CLIMADA, Sobol′ indices are generally well-suited for both uncertainty and sensitivity
analysis. For sampling the algorithm requires the use of the Sobol′ quasi-Monte-Carlo sequence (Sobol′, 2001), which provides

good rates of convergence when the number of input parameters is lower than $\sim 25$ (Lemieux, 2009). Sobol′ indices are obtained as the ratio of the marginal variances to the total variance of the output metric. In particular, the algorithm implemented in the *SALib* package allows to estimate the first-order, total-order and second-order indices (Saltelli, 2002). First-order indices measure the direct contribution to the output variance from individual input parameters. Total-order indices measure the overall contribution from an input parameter considering its direct effect, and its interactions with all the other input parameters. Second-order indices describe the sensitivity from all pairs of input parameters. In addition, the $95^{\text{th}}$ percentile confidence interval is provided for all indices. This allows to estimate whether the number $N$ of chosen samples was sufficient for both the uncertainty and sensitivity analysis. Note that in general the rate of convergence depends non-trivially on the number of input parameters, the probability distributions of the input parameters, the type of sensitivity index, and the sampling algorithm (Herman and Usher, 2017).

### 2.3.6 Sensitivity visualisation and statistics

The last step consists in analyzing and visualizing the obtained sensitivity indices. To this effect, a series of visualisation plots are provided, such as bar plots or sensitivity maps for first order indices, and correlation matrices for second order indices, as shown in Sects. 3.2.5 and 3.3.6. This step shows which input parameters' uncertainty is the driver of the uncertainty of each individual module output metric. This is useful to support model calibration and verification, to prioritize efforts for uncertainty reduction, and to inform robust decision-making.

## 3 Illustration with a case study on tropical cyclones storm surges in Vietnam

In the following we revisit a case study on tropical cyclone storm surges in Vietnam (Rana et al., 2021), and perform an uncertainty and sensitivity analysis on the risk assessment and adaptation options appraisal to illustrate the use of the CLIMADA unsequa module.

### 3.1 Case study description

We consider only the parts of the climate risk study by Rana et al. (2021) that modelled the impact of tropical cyclone storm surges in Vietnam in terms of number of affected people. The authors assessed the risk under present and future climate conditions, and performed an adaptation options appraisal by computing the benefits and costs for three physical adaptation measures – mangroves, sea dykes, and gabions. A more detailed recount of the case study is provided in Appendix C.

Below we showcase uncertainty and sensitivity analysis for the risk of storm surges in terms of affected people under present (2020) climate conditions in Sect. 3.2, and for the benefit and cost of the adaptation measure in 2050 considering the climate change Representative Concentration Pathways (RCP) 8.5 (Pachauri et al., 2015) in Sect. 3.3. The goal is to illustrate the use of the unsequa module, rather than to present a comprehensive uncertainty and sensitivity analysis for the case study. Thus, some of the uncertainties are defined in a stylised fashion by defining plausible distributions. A more in-depth analysis would

require the use of, e.g., an argument-based framework (Otth, 2021; Otth et al., 2022; Knüsel et al., 2020), and would be beyond the scope of this article.

For simplicity, hereafter (Rana et al., 2021) will be referred to as the *original* case study.

## 3.2 Risk assessment

The six steps of the uncertainty and sensitivity analysis (c.f. Fig. 1) are described in detail in the coming sections for the risk assessment of storm surges in Vietnam under present (2020) climate in terms of the number of affected people.

### 3.2.1 Input variables and parameters

We identified four main quantifiable uncertainty parameters which are summarized in the upper row of Table 1. As we remark above, the choice of the distribution of input parameters can substantially influence the results of the uncertainty and sensitivity analysis, and thus should ideally be based on background knowledge. The distribution chosen here are plausible, yet stylised, and should not be considered as general references for other case studies.

For the exposure, the total population is assumed to be subject to random sampling errors that are well captured by a normal distribution, and a maximum error of $\pm 10\%$ is assumed. Thus, the total population is scaled by a multiplicative input parameter $T$, distributed as a truncated Gaussian distribution, with clipping values $0.9, 1.1$, mean value $\mu = 1$ and variance $\sigma = 0.05$. For the population distribution, the original case study used the Gridded Population of the World dataset (Center for International Earth Science Information Network - CIESIN - Columbia University, 2018), which is available down to admin-3 levels. To account for uncertainties arising from the finite-resolution, we use the LitPop module from CLIMADA (Eberenz et al., 2020) to enhance the data with nightlight satellite imagery from the Black Marble annual composite of the VIIRS day-night band (Grayscale) at 15 arcsec resolution from the NASA Earth Observatory (Hillger et al., 2014), a common technique used to rescale population densities to higher resolutions (Anderson et al., 2014; Berger, 2020). In LitPop, the nightlight and population layers are raised to an exponent $m$ and $n$, respectively, before the disaggregation. Here, we vary the value of $m, n$ as a description of the uncertainty in the population distribution. In the original case study, $m$ is set to 0 and $n$ to 1. We consider the addition of the nightlight layer with $m \in (0, 0.5, 1)$, and vary the population layer with $n \in (0.75, 1, 1.25)$. The corresponding distributions are shown in Fig. A1. A higher value of $n$ emphasizes highly populated areas, a lower value the low populated areas. The corresponding input parameter $L$ represents all pairs of $(m, n)$.

For the hazard, we apply a bootstrapping technique, i.e., uniform re-sampling of the event set with replacement, in order to account for sample estimates uncertainties. Since the default Sobol$'$ global sampling algorithm requires repeated application of the same value of any given input parameter, here we define $H$ as the parameter that labels a configuration of the re-sampled events. Errors from the hazard modelling (c.f. Appendix C) are not further considered here. A more detailed study might want to explore further uncertainty sources, such as the windfield model, the hazard resolution or the random set generation algorithm.

Finally, for the impact function, we consider the uncertainty in the threshold of the original step-function that was used to estimate the number of people 'affected' (widely defined) by storm surges. In the original case study, the threshold was 1

| Risk assessment | | | | |
|---|---|---|---|---|
| Exposure | total value | T | truncated Gaussian multiplicative | clip:[0.9, 1.1] ; $\mu : 1, \sigma : 0.05$ |
| | spatial distribution | L | LitPop layers exponents | $m \in (0, 0.5, 1); n \in (0.75, 1, 1.25)$ |
| Hazard | event set bootstrapping | H | re-sampling the event set with replacement | |
| Impact function | threshold shift | S | uniform range | [0.5m, 3.0m] |

| Adaptation options appraisal | | | | |
|---|---|---|---|---|
| Exposure | total value | T | truncated Gaussian multiplicative | clip:[0.9, 1.1] ; $\mu : 1, \sigma : 0.05$ |
| | spatial distribution | L | LitPop layers exponents | $m \in (0, 0.5, 1); n \in (0.75, 1, 1.25)$ |
| Hazard | event set bootstrapping | H | re-sampling the event set with replacement | |
| Impact function | threshold shift | S | uniform range | [0.5m, 3.0m] |
| Population growth | growth rate | G | uniform range (case study value: 1.13) | [1.10, 1.16] |
| Climate change | hazard intensity | I | uniform range multiplicative | [0.9, 1.1] |
| | hazard frequency | F | uniform range multiplicative | [0.5, 2.0] |
| Cost of all adaptation measures | total cost | C | uniform range multiplicative | [0.5, 2.0] |

**Table 1.** Summary of the input parameter distributions. The input parameters $T, L, G$ characterize the uncertainty in the exposure (people), $H, I, F$ in the hazard (storm surge), $S$ in the impact function (vulnerability), and $C$ in the adaptation measures (mangroves, sea dykes, gabions). The parameters $T, L, H, S$ are needed for risk assessment (c.f. Sect. 3.2.1), and the parameters $T, L, H, S, G, I, F, C$ are needed for adaptation options appraisal (c.f. Sect. 3.3.1).

meter, with $0\%$ affected people below, and $100\%$ affected people above. We consider a threshold shift between 0.5m and 3m. This extends a range examined in a study of human displacement due to river flooding (there from 0.5-2m) (Kam et al., 2021), in order to more widely explore uncertainty related to resolution of the population and topography. This distribution does not examine a specific impact, but rather how the total number of people 'affected' varies based on different thresholds used to define 'affected'. The resulting range of impact function is shown in Fig. A2.

### 3.2.2 Samples

For the sampling we use the default Sobol′ sampling algorithm (Sobol′, 2001; Saltelli and Annoni, 2010) to generate a total of 10240 samples as shown in Fig. A3.

### 3.2.3 Model output

For each of the samples $n$, the full impact matrix $I_n$ is obtained and saved for later use. From the impact matrix we furthermore
compute several risk metrics for each sample: the average annual impact aggregated over all exposure points, the aggregated risk at returns periods of $5, 10, 20, 50, 100, 250$ years, the impact at each exposure point, as well as the aggregated impact for each event (for details c.f. Aznar-Siguan and Bresch, 2019).

### 3.2.4 Uncertainty visualisation and statistics

In the following, we concentrate on the analysis of the full uncertainty distribution of various risk metrics. For convenience, the original case study value, the uncertainty mean value and standard deviation are also reported. But, as we shall see below, focusing only on these numbers would provide a limited picture.

The full uncertainty distribution for each of the return periods, as well as the exceedance frequency curve are shown in Fig. 2. First, we remark that the original case study exceedance frequency curve, shown in Fig. 2 (b), is close to the median percentile, while the upper and lower $95^{th}$ percentile of the uncertainty are roughly $+40\%$ and $-60\%$ compared to the median, respectively. Second, the distribution of uncertainty for each return period separately, shown in Fig. 2 (a), is in fact bi-modal, in particular for shorter return periods. The original case study values for the lower return periods are all among the higher mode. Third, the distribution of the average annual impact aggregated over all exposure points, shown in Fig. 2 (c), is also bi-modal, with the original case studies lying in the mode with larger impacts. The mean number of affected people is $1.42$M with a variance of $\pm 1.03$M, which is compatible with, but lower than the original case study value of $1.94$M.

The bi-modal form of the impact uncertainty distribution is interesting, as one could rather expect statistical white or colored noise (e.g., Gaussian or power-law distributions). As a proof-of-consistency that this is not due to a computational setup error, we verified that the distribution of the total asset value, shown in Fig. 2 (d), aligns with the parametrization of the exposure uncertainty (c.f. Table 1). For a better understanding of the obtained uncertainty distributions, and in particular understand the bi-modality, let us continue with the sensitivity analysis.

### 3.2.5 Sensitivity indices

Ideally, we should choose the sensitivity method best suited for the data at hand. In our case, the uncertainty distribution is strongly asymmetric (c.f., Fig. 2), thus a density-based approach would be best (Pianosi and Wagener, 2015; Borgonovo, 2007; Plischke et al., 2013). However, this would require generating a new set of samples, and for the purpose of this demonstration we used the unsequa default variance-based Sobol method. Note that despite the questionable use of variances to characterize sensitivity for multi-modal uncertainty distributions, the derived indices prove useful to better understand the results from the case study at hand.

We thus computed the total-order and the second-order Sobol$'$ indices (Sobol$'$, 2001) for all the input parameters $T, L, H, S$. We obtained the sensitivity indices for all the risk metrics shown in Fig. 2: average annual impact aggregated (aai_agg), impact for return periods of $5, 10, 20, 50, 100, 250$ years (rp5, rp10, rp20, rp50, rp100, rp250) and in addition for the average annual impact at each exposure point.

### 3.2.6 Sensitivity visualisation and statistics

As shown in Fig. 3 (a), for the average annual impact aggregated, the largest total-order sensitivity index is for the impact function threshold shift with $ST_S(\text{aai\_agg}) \approx 0.95$. This indicates that the uncertainty in the impact function threshold shift $S$ is the main driver of the uncertainty. Thus, to understand the bi-modality of the uncertainty distribution (c.f. Fig. 2 (c)), we have

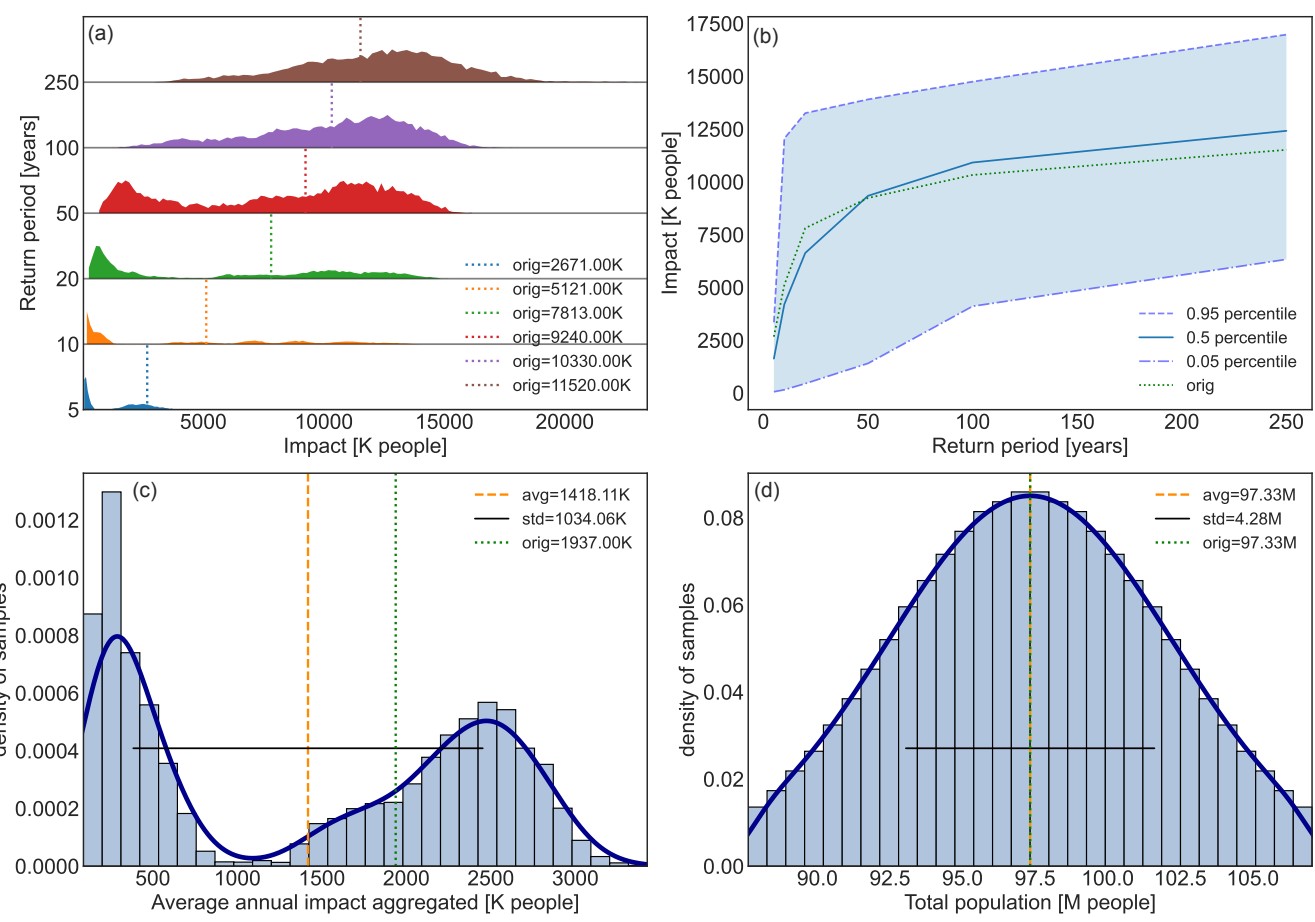

**Figure 2.** Uncertainty distribution for storm surge risk in terms of affected people in Vietnam for present climate conditions (2020). (a) Full range of the uncertainty distribution of impacted people for each return period $(5, 10, 20, 50, 100, 250$ years), and value in the original study (dotted vertical lines); (b) Impact exceedance frequency curve shown for the original case study results (green dotted line), the median percentile (solid blue line), $5^{th}$ percentile (dash-dotted blue line), and $95^{th}$ percentile (dashed blue line); (c) Distribution of annual average impact aggregated over all exposure points (histogram bars) and (d) Distribution of the uncertainty of the total population, i.e., the total exposure value, (histogram bars), both including the average value (dashed orange vertical line), original case study result (dotted green vertical line), standard deviation (solid black horizontal line), and kernel density estimation fit to guide the eye (solid dark blue line). The impacts are expressed in thousands (K) or millions (M) affected people.

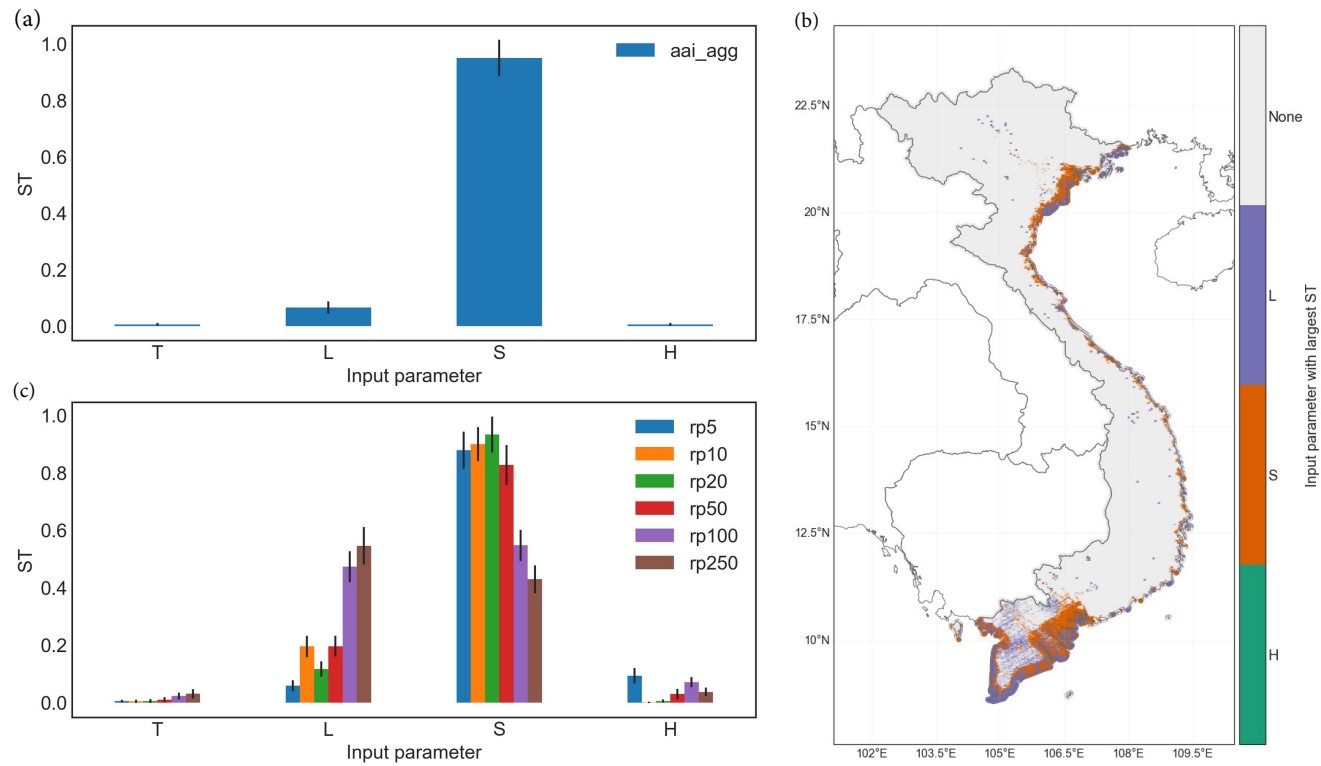

**Figure 3.** Total order Sobol$'$ sensitivity indices (ST) for storm surge risk for people in Vietnam for present climate conditions (2020). (a) Results shown for the annual average impact aggregated over all exposure points (aai_agg); (b) map of the largest sensitivity index at each exposure point. The category 'None' refers to areas with vanishing risk. (c) Sensitivity results for risk estimate over return periods (rp) $5, 10, 20, 50, 100, 250$ years. The input parameters (c.f., Table 1) are T: total population, $L$: population distribution, S: impact function threshold shift, and $H$: hazard events bootstrapping. The vertical black bars in (a)(b) indicate the $95^{th}$ percentile confidence interval.

to better understand the relation between $S$ and the model output. Note that there are no strong interactions between the input parameter uncertainties as all second-order sensitivity indices S2 $\approx 0$ (c.f., Fig. A6). Thus, it is reasonable to assume that the bi-modality of the distribution comes directly from $S$ and not from correlation with other input parameters. We further remark that the $95^{th}$ percentile confidence intervals of the sensitivity indices (indicated with vertical black bars in Fig. 3) are much smaller than the difference between the sensitivity indices. We thus conclude that the number of samples was sufficient for a

reasonable convergence of the uncertainty and sensitivity sampling algorithm.

A further analysis of the average annual impact aggregated value in function of the impact function threshold shift $S$ reveals a discontinuity at a value of $S_d \sim 1.85m$ as shown in Fig. A5 (a). Hence, the bi-modality of the uncertainty distributions (c.f., Fig. 2) is indeed due to the uncertainty input parameter $S$ of the impact function, but does not explain the root cause. Further understanding is obtained from studying the storm surge footprint used in the original case study. Plotting the storm surge

intensity of all events at each location with values ordered from smallest to largest, we find a discontinuity and plateau around

$1.85m$, as shown in Fig. A5 (b). This is precisely the value corresponding to the threshold shift at which the annual average impact is discontinuous. Thus, the bi-modality of the uncertainty distributions, while caused by uncertainty in the impact function roots in the modeling of the storm surge hazard footprints. Further research beyond the scope of this paper would be need to understand whether this value of 1.85m has a physical origin (e.g., landscape features or protection standards), or is due to a modelling artefact. However, despite the discontinuity, the patterns are as expected: an impact function with a step at 0.5m results in many more people being classified as affected than when the steps is at 3m (in the latter case, only particularly large storm surges would results in people being affected). For planning purposes, the lower end of this impact function shift is most relevant – even 0.5m depth of storm surge can be dangerous for people - so the higher mode of the distribution in Fig. 2 is most relevant.

At last, in Fig. 3 (b) the largest sensitivity index for the average annual impact at each exposure point is reported on a map. In the highly populated regions around Ho Chi Minh city (South Vietnam) and Haiphong (North Vietnam), the largest index is $S$ in accordance with the sensitivity of the average annual impact aggregated over all of Vietnam. However, in less densely populated areas, such as the larger Mekong delta (South Vietnam), the outcome is more sensitivity to the population distribution $L$. Furthermore, while for shorter return periods, the largest total-order sensitivity index is the impact function threshold shift $ST_S$, for longer return periods the sensitivity to the population distribution $ST_L$ gets larger as shown in Fig. 3 (c). This might be because stronger events with large return periods consistently have larger intensities than the maximum threshold shift of $3m$. Together, these results hint to potentially hidden high impact events in unexpected areas (e.g., a large storm surge in the less densely populated southern tip of Vietnam could affect a large number of people).

## 3.3 Adaptation options appraisal

We focus on the appraisal of the three adaptation measures mangroves, sea dykes, and gabions to reduce the number of people affected by storm surges assuming the high-emission climate change scenario RCP8.5. We consider the time frame $2020-2050$ as in the original case study.

### 3.3.1 Input variables and parameters

We identified four additional quantifiable uncertainty input parameters for the adaptation options appraisal compared to the risk assessment study (c.f. Sect. 3.2.1) that are summarized in the bottom row of Table 1. For the exposure, the growth rate of the population from 2020 to 2050 was estimated at $13\%$ in the original case study based on data from the United Nations (Nations, 2019). We here assume a growth rate $G$ uniformly sampled between $10\%$ and $16\%$. For the hazard, the original case study used the parameters from Knutson et al. (2015) to scale the intensity and frequency of the events considering the climate change scenario RCP8.5 from 2020 to 2050 (see Appendix C for more details) . This method is subject to large uncertainties (see e.g. Knüsel et al., 2020) and we thus scale the intensity and frequency with parameters $I$ and $F$ uniformly sampled from $[0.9, 1.1]$ and $[0.5, 2]$, respectively. Finally, the cost of the adaptation measures is assumed to vary by a multiplicative factor $C$, sampled uniformly between $[0.5, 2]$.

### 3.3.2 Samples

For the sampling we use the default Sobol$'$ sampling algorithm to generate a total of $N = 18432$ samples. Owing to the larger amount of input parameters, the total number of samples is larger than for the risk assessment (c.f. Sect. 3.2.2). The drawn samples are shown in Fig. A4.

### 3.3.3 Model output

For each of the samples, we obtained the cumulative output metrics over the whole time period $2020 - 2050$. In particular, we obtained the total risk without adaptation measure, the benefits (averted risk) for each adaptation measure, and the cost of each adaptation measure (for details see Bresch and Aznar-Siguan, 2021). One can then compare the cost-benefit ratios, i.e. the cost in dollars per reduced number of affected people, for each of the adaptation measure including model uncertainties.

### 3.3.4 Uncertainty visualisation and statistics

The uncertainty for the cumulative, total average annual risk from storm surges aggregated over all exposure points is shown in Fig. 4 (d). The distribution is bi-modal, which can be traced back to the storm surge model as explained in Sect. 3.2.3. The original case study value is located in the larger mode, similarly to the average annual risk in 2020 as discussed in Sect. 3.2.3. This bi-modality translates to the uncertainty in the benefit (total averted risk) for the adaptation measure sea dykes, Fig. 4 (b), but not to the adaptation measures mangroves and gabions, Fig. 4 (a) and (c). Rather, the latter show a heavy-tail uncertainty distribution. Furthermore, the uncertainty analysis of the ratio of the cost to the benefits for each adaptation measure indicates that, contrary to the original case study, the sea dykes might in fact be the *least* (instead of the most) cost-efficient adaptation measure (see Fig. A7 (a)-(c)). Note that expressing the cost-efficiency of an adaptation measure in terms of reduced number of affected people for each invested dollar presents ethical challenges as will be discussed in more detail in Sect. 4.

### 3.3.5 Sensitivity indices

We use the same method as for the risk assessment to compute the total-order ST and the second-order S2 Sobol$'$ indices (Sobol$'$, 2001) for all the input parameters $T, L, G, H, F, I, S, C$ (c.f. Table 1). We obtain the sensitivity indices for all the metrics shown in Fig. 4 and Fig. A7, i.e., the total risk as well as the benefits and cost-benefit ratios for all adaptation measures.

### 3.3.6 Sensitivity visualisation and statistics

The total risk without adaptation measure is most sensitive to the impact function threshold shift $S$ with $\mathrm{ST}_S(\text{total risk}) \approx 0.85$ as shown in Fig. 5 (b). In addition, the sensitivity to the storm surge frequency changes $ST_F(\text{total risk}) \approx 0.18$ is significantly larger than the sensitivity to the intensity changes $S_I(\text{total risk}) \approx 0.02$. This could be a consequence of the choice to use a step-function to model the vulnerability.

The uncertainty of the benefits for all adaptation measures are most sensitive to the impact function threshold shift, with $\mathrm{ST}_S^{\mathrm{mangroves}}(\text{benefit}) \approx \mathrm{ST}_S^{\mathrm{gabions}}(\text{benefit}) \approx 0.85$, and $\mathrm{ST}_S^{\text{sea dykes}}(\text{benefit}) \approx 0.75$ as shown in Fig. 5 (a). This is consistent with

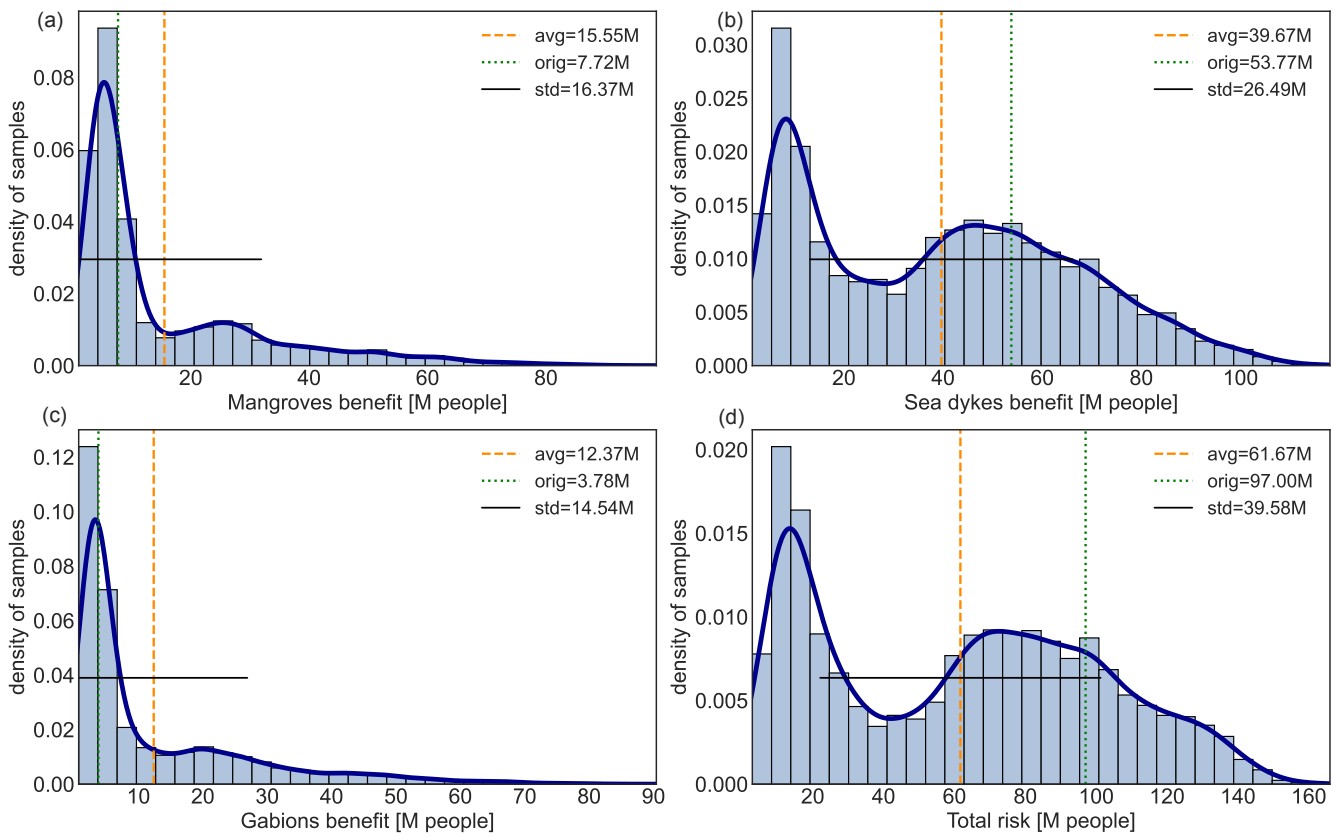

**Figure 4.** Uncertainty distribution (histogram bars) for benefits (averted risk) from the adaptation measures (a) mangroves, (b) sea dykes, (c) gabions, and (d) the total risk without adaptation. Benefits and total risk are cumulative over the time period $2020 - 2050$ for the climate scenario RCP8.5. Dotted green vertical lines indicate the case study value, dashed orange vertical lines the average benefit over the uncertainty distribution, the solid black horizontal line shows the standard deviation, and the solid dark blue line the kernel density estimation fit to guide the eye. The benefits and total risk are expressed in millions (M) of affected people.

the sensitivity of the risk in 2020 (c.f., Fig. 3). In addition, there is some sensitivity to the people distribution $L$, and to the uncertainty in the climate change input parameters $I$ and $F$. Note, however, that $\mathrm{ST}_I^{\text{sea dykes}}(\text{benefit}) \approx 0$, i.e., the uncertainty of the benefits from the adaptation measure sea dykes is almost not sensitive to the intensity changes uncertainty, while for both mangroves and gabions it is. This could be because sea dykes are parametrized to reduce the storm surge level by 2 meters, which is above the $S_d = 1.85$ meters identified in Sect. 3.2.3 as critical for the surge modelling, while gabions and mangroves are parametrized to provide a reduction of $0.5$ meter which is below (c.f. Appendix C and Rana et al. (2021)). Thus, a change in the hazard frequency and the population distribution patterns will result in a stronger variation of the benefits for sea dykes because fewer, but stronger events contribute to the remaining risk each year.

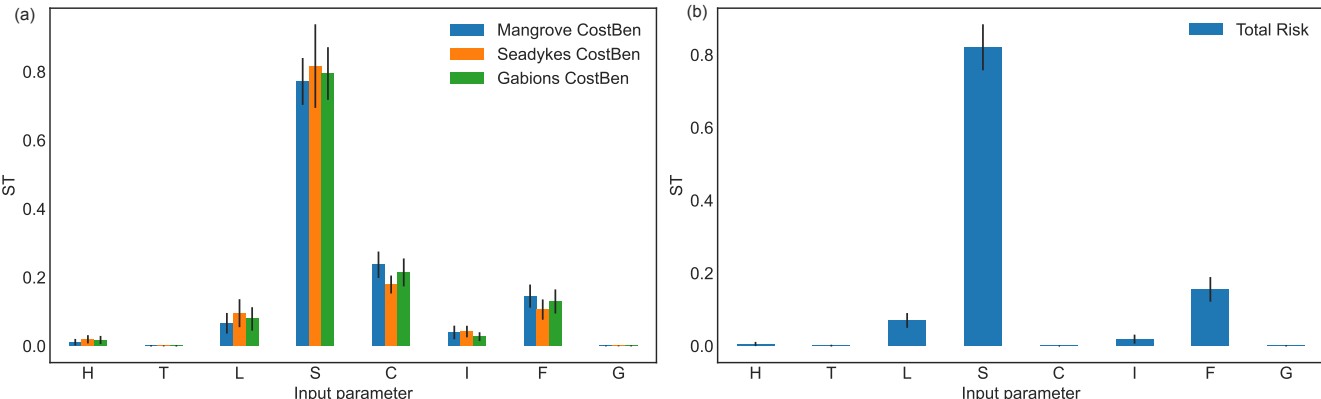

**Figure 5.** Total order Sobol′ sensitivity indices (ST) for the uncertainty of (a) storm surge adaptation options benefits for mangroves, sea dykes and gabions and of (b) the total risk without adaptation, for the time period $2020 - 2050$ under the climate scenario RCP8.5. The input parameters (c.f., Table 1) are $H$: hazard events bootstrapping, T: total population, $L$: population distribution, S: impact function intensity threshold shift, C: cost of adaptation options, I: hazard intensity change, F: hazard frequency change, and G: population growth. The vertical solid black bars indicate the 95[th] percentile confidence interval.

Note that the $95^{th}$ percentile confidence intervals of the sensitivity indices (indicated with vertical black bars in Fig. 5) are much smaller than the difference between the sensitivity indices. We thus conclude that the number of samples was sufficient for a reasonable convergence of the uncertainty and sensitivity sampling algorithm.

### 3.4 Summary of the case study

The original case study intended to serve as a blueprint for future analyses of other world regions with limited data availability, and thus focused on the application of established research tools to provide insights into natural hazard risks and potential benefits of adaptation options (Rana et al., 2021). In view of limited observational data for impacts from tropical cyclones, the results of the study should have been subject to considerable uncertainty. The need of uncertainty and sensitivity analysis was identified within the original study, but deemed out of scope. This was in part due to the absence of a comprehensive and

easily applicable scheme, now resolved with the uncertainty and sensitivity quantification module presented here. In addition, a full-fledged uncertainty and sensitivity analysis leads to a large amount of additional data to process. Indeed, the results shown in this section considered only a small subset of the original case study, which, among others, also considered the impact of tropical cyclone wind gusts, and the impact of wind and surge on physical assets in dollars. Nevertheless, the benefits of an uncertainty and sensitivity analysis are manifest. On the one hand, it provides a much more comprehensive picture on risk from

storm surges and the benefits of identified adaptation measures. On the other hand, it allows to identify the main shortcomings of the model, which is needed to focus modelling improvement efforts and to understand the limitations of the obtained results. Even when used in the context of studies such as ECA, which are bound by time and money, this is useful to improve the confidence in, and transparency of the outcomes, and allows for model improvements from study to study. In this section for

instance, it was conclusively shown that in order to improve the impact modelling, one should focus on the storm surge model, among other aspects. Furthermore, the analysis showed that urban and rural regions might not be equally well-represented by the model.

## 4  Discussion and outlook

In this paper we described the unsequa module for uncertainty and sensitivity analysis recently added to the climate risk model CLIMADA. We highlighted its ease of use with an application to a previous case study assessing risks from tropical storm surges to people in Vietnam and appraising local adaptation options. We showed that only providing percentile information without the full distributions can be misleading, and that uncertainty analysis without sensitivity analysis does not provide a thorough picture of uncertainty (Saltelli and Annoni, 2010). The example showed the vital role played by uncertainty and sensitivity analysis in not only producing better and more transparent modelling data, but also providing a more comprehensive context to quantitative results in order to better support robust decision making (Wilby and Dessai, 2010). This expansion of the CLIMADA platform allows for risk assessment and options appraisal including quantification of uncertainties in a modular form and occasionally bespoke fashion (Hinkel and Bisaro, 2016), yet with the high re-usability of common functionalities to foster usage in interdisciplinary studies (Souvignet et al., 2016) and international collaboration. Further, the presented approach can be used to inform the development of story-lines (Shepherd et al., 2018; Ciullo et al., 2021) and climate adaptation narratives (Krauß and Bremer, 2020).

The illustrative case study in this paper was run on a computing cluster. However, many potential users will not have access to such computational resources. Nonetheless, meaningful uncertainty and sensitivity  analysis can be conducted only on a single computer, by for instance reducing resolution, sample size, or the number of uncertainty input parameters. For example, the illustrative case study in the paper could be run reasonably on a typical laptop by reducing the resolution to 150arcsec. By doing so, it is not possible to explore all possible nuances, but one can still get a big-picture view of where key areas of uncertainty and sensitivity may lie.

While we showed that quantitative uncertainty and sensitivity is a significant step to improve the information value of climate risk models, we stress that not all uncertainties can be described with the shown method (see e.g., Appendix D for a discussion on event uncertainty). Indeed, only the uncertainty of those input parameters that are varied can be quantified, and even for these input parameters, defining the probability distribution is subject to strong uncertainties, often being based only on educated guesses. Yet, the choice of probability distribution can have a strong impact on the resulting model output distribution and sensitivity (Paleari and Confalonieri, 2016; Otth, 2021; Otth et al., 2022). In addition, it is often not evident how to perturb the input variables, since one does not always have access to the underlying generating model, and it is otherwise difficult to define physically consistent statistical perturbations of geospatial data. Moreover, there is a large part of climate risk models uncertainty that is not even in principle quantifiable (Beven et al., 2018b; Knüsel et al., 2020). When building a climate risk model, a number of things must be specified, such as the model type, the algorithmic structure, the input data, the resolution, the calibration and validation data etc. These choices are often not made based on solid knowledge (Knüsel, 2020). One particular

type of uncertainties, with which modellers are less familiar are *normative* uncertainties that arise from value-driven modelling choices (Bradley and Drechsler, 2014; Bradley and Steele, 2015) that are particularly relevant when the climate risk analysis is carried out to support decisions and options appraisal. Normative uncertainties are rarely identified in common modeling practice (Bradley and Drechsler, 2014; Bradley and Steele, 2015; Moeller, 2016; Mayer et al., 2017). In most cases, these uncertainties can hardly be quantified and, therefore, they need to be addressed via methods such as argument analysis (Knüsel et al., 2020), the NUSAP methodology (Funtowicz and Ravetz, 1990) or sensitivity auditing (Saltelli et al., 2013). In some other cases, e.g., the decision regarding the value of a discount rate, normative uncertainties can be quantified, and quantitative analyses can highlight the effects of varying modeling choices on the decision outcomes. A complementary study to this paper proposes a methodological framework for a broader assessment of uncertainties for decision processes with CLIMADA as the climate risk model, including both conceptual and quantitative approaches (Otth, 2021; Otth et al., 2022).

If a climate risk modeller conducts an uncertainty and sensitivity analysis, either by using the CLIMADA module published here, or by implementing a similar analysis in another modelling framework, the next question is: what should be done with the results? We suggest two main areas that could benefit from such analyses. First, within the field, the more that uncertainty and sensitivity analyses become standard practice, the more these analyses will enhance transparency of studies among climate risk modellers. This can help to focus related research on areas that can provide better understanding of the parameters, or on modelling choices that are most influential on model outputs. Second, for decision-makers and other users of climate risk modelling, uncertainty and sensitivity analysis has the potential to lead to better-informed decisions on climate adaptation. Several methods exist to inclusion into quantitative decision making analysis (Hyde, 2006). Certainly, the numerical and graphical outputs of the module published here, or outputs from similar analyses, are far too technical to directly hand over as-is to decision-makers and other users (unless the user is a risk analyst already versed in uncertainty and sensitivity analyses). Rather, the results of uncertainty and sensitivity analysis can inform discussions between climate risk modellers and decision-makers about how best to refine and interpret model results. It is especially important to reflect additionally on uncertainties that lie outside the model and thus were not analysed in the quantitative uncertainty and sensitivity analysis (Otth, 2021; Otth et al., 2022). Further research and reflective practice can focus on how to most effectively achieve this.

In future iterations, uncertainty analysis in CLIMADA could be extended with for instance the addition of surrogate models to reduce the computational costs and allow for the testing of larger number of input parameter with larger number of samples for models at higher resolution. Overall, we hope that the simplicity of use of the presented unsequa module will motivate modellers to include uncertainty and sensitivity analysis as a natural part of climate risk modelling. Finally, we caution that numbers even with elaborate error bars and distributions can give a false sense of accuracy (Hinkel and Bisaro, 2016; Katzav et al., 2021) and that modellers should remember to reflect on the wider, non-quantifiable uncertainties, unknowns and normative choices of their models.

*Code and data availability.* CLIMADA is openly available at GitHub https://github.com/CLIMADA-project/climada_python, (Kropf et al., 2022) under the GNU GPL license (GNU operating system, 2007). The documentation is hosted on Read the Docs https://climada-python.

readthedocs.io/en/stable/ and includes a link to the interactive tutorial of CLIMADA. In this publication, CLIMADA v3.1.0, deposited on Zenodo (Kropf et al., 2022) was used. The scripts to reproduce the data from the case study (Rana et al., 2022) is available from https://github.com/arunranain/climada_tc_vietnam.

*Data availability.* All data has been generated using CLIMADA (the LitPop exposures, the impact function, the storm surge hazard, the adaptation measures, all impact and cost-benefit values, the uncertainty distributions and the sensitivity indices). Detailed tutorials are avail-

535 able at https://climada-python.readthedocs.io/en/v3.1.1/ (version 3.1.) and at https://climada-python.readthedocs.io/en/stable/ (latest stable version). For generating the storm surge hazard in 2020 and 2050, a digital elevation model (DEM) was used which is not included in CLIMADA. The hazards have been made available under the DOI https://doi.org/10.3929/ethz-b-000566528. The scripts to reproduce the case study (Rana et al., 2022) data (impact function, adaptation measures, hazard, exposures, impact, cost-benefit) are available at https://github.com/arunranain/climada_tc_vietnam.

## Appendix A: Sampling algorithms

CLIMADA imports the quasi Monte-Carlo sampling algorithms from the *SALib* Python package (Herman and Usher, 2017). Thus all sampling algorithms from this package are directly available to the user within the new module. These algorithms are all at least implemented for uniform distribution $p^u$ over $[0, 1]$. In order to accommodate for any input parameter distributions, the CLIMADA module uses the percent point function (ppf) $Q$ (also called inverse cumulative distribution, percentiles or quantile function) of the target probability density distribution. For example, in order to obtain a sample of $N$ Gaussian distributed $p^G$ values, one first samples $X^u = x_1, x_2, \ldots, x_N$ values uniformly from $[0, 1]$, and then apply the ppf of the Gaussian distribution $Q^G$,

$$X^u = x(1), x(2), \ldots, x(N) \to X^G = Q^G(x(1)), Q^G(x(2)), \ldots, Q^G(x_(N)). \tag{A1}$$

## Appendix B: Helper methods

In the unsequa module, a number of helper methods exist to parameterize a few common uncertainty parameter distributions for the main input variables exposure, impact functions, hazard and measures, as summarized in table B1. *These helper methods are for the convenience of the users only. Any other uncertainty parameter can be introduced, and any other uncertainty parameter distributions (discrete, continuous, multi-dimensional, etc.) can be defined by the user if needed.* The user could thus for instance write a wrapper function around an existing dynamical hazard model which outputs a CLIMADA hazard

object, and define the input factors of said dynamical model as uncertainty parameters.

For risk assessment, the impact at an exposure location $x$ for an event $\epsilon$ is defined (Aznar-Siguan and Bresch, 2019) as

$$I_{x,\epsilon} = f_x(h_\epsilon(\tilde{x})) \cdot a_\epsilon(\tilde{x}) \cdot v(x) \tag{B1}$$

| Input variable | Input parameter | Distribution | Equation |
|---|---|---|---|
| Exposure | Total value $T$ | Uniform | $v(x) \cdot T$ |
| | Value noise $N$ | Multiplicative Gaussian noise on each value* | $v(x) + \delta_x^N$ |
| | List members $L$ | Uniform choice** | $v \to v_L$ |
| Hazard | Intensity $I$ | Uniform | $h_{\epsilon,x} \cdot I$ |
| | Fraction $A$ | Uniform | $a_{\epsilon,x} \cdot A$ |
| | Frequency $F$ | Uniform | $\nu_\epsilon \cdot F$ |
| | Resampling $E$ | Re-sampling with replacement*** | $\{h\}_\epsilon \to \{h\}_{\epsilon E}$ |
| | List members $K$ | Uniform choice** | $h \to h_K$ |
| Impact function | Intensity $i$ | Uniform | $f(x) \to f(x+i)$ |
| | MDD $D$ | Uniform | $f(x) \to f(x) + D$ |
| | List members $F$ | Uniform choice** | $f \to f_F$ |

**Table B1.** Summary of available helper methods to define uncertainty parameters distributions for the main input variables of CLIMADA for risk assessment. For all distribution the parameters can be set by the user (e.g., the mean and variance of a Gaussian distribution are free parameters). * The additive noise terms $\delta_x^N$ are all independently, identically sampled from the same truncated Gaussian distribution. The input parameter $N$ labels the noise realizations (one realization consists in one value $\delta_x$ for all locations $x$). ** The user can define a list of exposures, hazards of impact functions to uniformly choose from. For instance, a list of exposures with different resolutions, or a series of LitPop exposures with different exponents can used as shown in Fig. A1. Another example would be to define a pre-computed list of Hazards obtained from a dynamical model (e.g., a flood model) for different dynamical model input factors, or use a list of Hazards obtained from different data sources. Analogously, a list of impact functions obtained, for example with different calibration methods, could be used. *** Events are sampled with uniform probability and with replacement. The size of the resampled subsets is a free parameter. For instance, size equal to one would correspond to considering single events, and size equal to the total number of events would correspond to bootstrapping. The input parameter $E$ labels one set of re-sampled events. *These helper methods are for the convenience of the users only. Any other uncertainty parameter distributions (discrete, continuous, multi-dimensional, etc.) can be defined by the user if needed.*

where $f_x$ is the impact function for the exposure at location $x$, $h_\epsilon(\tilde{x})$ and $a_\epsilon(\tilde{x})$ are the hazard intensity and fraction of event $\epsilon$ at the location $\tilde{x}$ closest to $x$, and $v(x)$ is the value of the exposure at location $x$. Considering all locations and all events defines the impact matrix $I$. All further risk metrics, such as the average annual impact aggregated, are derived from $I$ and the annual frequency $\nu_\epsilon$ of each hazard event. The helper methods are defined to describe generic uncertainties on the input variables $v$, $f$, $\nu$, $h$ and $a$.

For the adaptation option appraisal, the measures are represented as a modification of the exposure, impact functions, or hazard, at a given cost. Thus, all the helper methods for the exposure, impact functions and hazard defined in Table B1 can be used for the measures uncertainty. In addition, the discount rate used to properly consider future economic risks can be defined. Thus, two additional helper methods for uncertainty in the cost $c$ and the discount rate $d$ are defined in Table B2.

| Input variable | Input parameter | Distribution | Equation |
|---|---|---|---|
| Measures | Cost $C$ | Uniform | $c \cdot C$ |
| Discount rates | Rate $D$ | Uniform choice [*] | $D$ |

**Table B2.** Summary of available helper methods to define uncertainty parameters distributions for the additional input variables of CLIMADA required for adaptation option appraisal. For all distribution the parameters can be set by the user (e.g., the bounds of the uniform distributions are free parameters). [*] The discount rate value is sampled uniformly from a list of values. *These helper methods are for the convenience of the users only. Any other uncertainty parameter distributions (discrete, continuous, multi-dimensional, etc.) can be defined by the user if needed.*

## Appendix C: Case study details

In the Vietnam case study Rana et al. (2021), probabilistic tropical cyclones hazard datasets for storm surges were created for the period of $1980 - 2020$, based on 269 historical, land-falling events recorded in the global International Best Track Archive for Climate Stewardship (IBTrACS) (Knapp et al., 2010). These historical tropical cyclone records were extended using a random walk algorithm to produce 99 probabilistic tracks for each record, yielding a large set of synthetic events (Kleppek et al., 2008; Gettelman et al., 2017; Aznar-Siguan and Bresch, 2019). A $2D$ windfield was calculated for each track using the wind model after Holland (2008). The surge hazard dataset (flood depth) is derived from wind intensity with a linear relationship that modifies the water level according to the local elevation and distance to the coastal line as further described in Rana et al. (2021). Future climate hazard sets were created for two Relative Concentration Pathways (RCP) (Pachauri et al., 2015), RCP6.0 and RCP8.5, based on parametric estimates. For each storm, the intensity and frequency where homogeneously shifted by a multiplicative constant derived from (Knutson et al., 2015) based on the storm's Saffir-Simpson category.

The spatial distribution of population was obtained from the LitPop module in CLIMADA at a resolution of 1 km and using the population census data only, i.e., $m = 0, n = 1$ (Eberenz et al., 2020). For the future scenario, a total population growth is estimated to amount to $13\%$ until 2050 based on estimates from the United Nations (2019). The impact function for the effect of storm surges on population was created in consultation with experts in the field; all people are considered affected at $1m$ water depth (Rana et al., 2021). Benefit and cost information on the three adaptation measures (sea dykes, gabions, mangroves) are given in Table 3. in Rana et al. (2021).

## Appendix D: Event uncertainty

As stated in Sect. 4, not all quantifiable uncertainties are described with the quasi Monte-Carlo method described in this paper. For instance, the uncertainty in climate risk arising from the inherent stochasticity of weather events can be directly described without using the unsequa module. In CLIMADA, this variability is directly modelled by considering the hazard to be a probabilistic set of events, i.e., intensity maps with associated frequencies (Aznar-Siguan and Bresch, 2019). Computing the

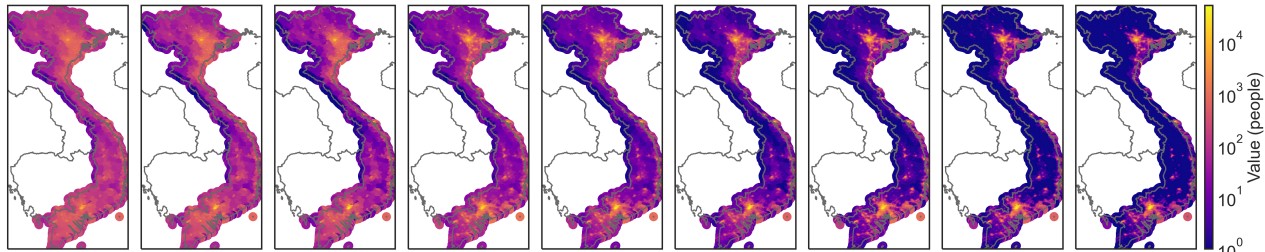

**Figure A1.** Population distribution obtained by combining population density layer and nightlight satellite imagery (cf. Litpop method, Eberenz et al., 2020) for all combinations of the nightlight and population exponents m and n considered in the uncertainty analysis (c.f. Table 1). From left to right, $m, n = (0, 0.75); (0, 1); (0, 1.25); (0.5, 0.75); (0.5, 1); (0.5, 1.25); (1, 0.75); (1, 1); (1, 1.25)$, with $(0, 1)$ the original case study value.

risk from the hazard amounts to computing the risk for each event in the set, which results in a probabilistic risk distribution. The event risk distribution expresses the fact that we do not know when a particular natural hazard event will happen, and qualifies as aleatory uncertainty (Uusitalo et al., 2015; Ghanem et al., 2017). One can compute statistical values, such as the mean or standard deviation, or consider the full distribution over the event set as shown in Fig. A8 (a) for the original case study risk. There is no need for an extra sampling (and use of the unsequa module) to determine this uncertainty, as this is part of the modelling of the hazard. Note however, that this variability is itself subject to modelling uncertainty. The distribution of risk obtained over all events and all input parameter samples, as shown in Fig. A8 (b), can then be seen as an estimate of the weather risk variability, including additional uncertainties.

Note that in general, global uncertainty and sensitivity analysis as discussed in this paper apply only to deterministic computer codes, i.e., models for which a specific set of input values always results in the same output (Saltelli, 2008; Marrel et al., 2012). CLIMADA is such a deterministic computer code. In order to describe truly stochastic models on would have to use other techniques, for instance which allow to take into account correlations between input parameters, or which are directly built for probabilistic computer codes (Ehre et al., 2020; Étoré et al., 2020; Zhu and Sudret, 2021).

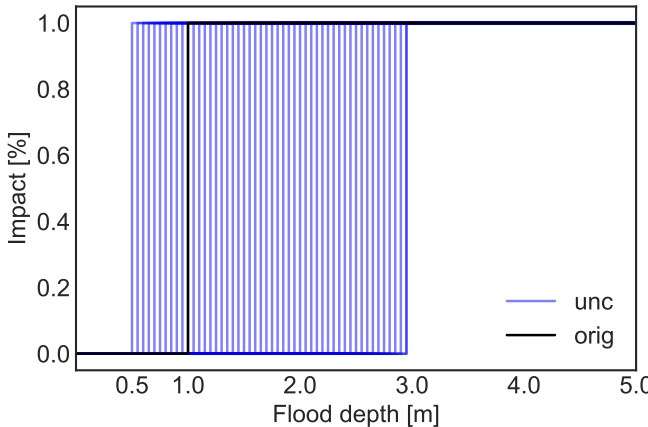

**Figure A2.** Impact function uncertainty, with a threshold shift of the flood depth above which all people are affected varying between 0.5m and 3m (c.f., Table 1). The original impact function is given in black.

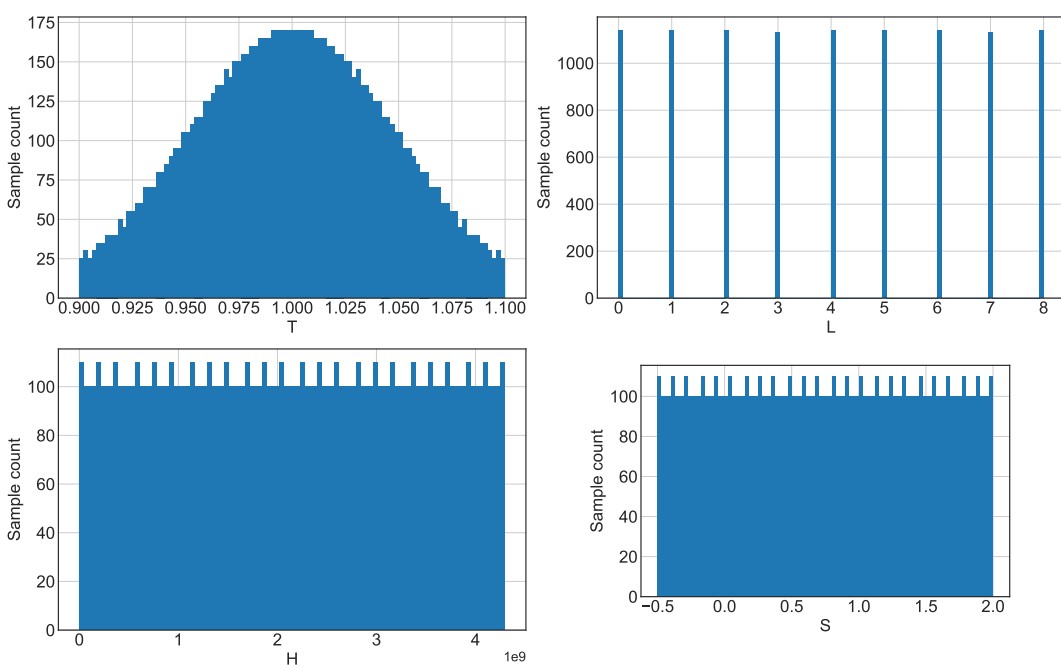

**Figure A3.** Samples for the uncertainty analysis of the risk assessment in Sect. 3.2.2 for the input parameters drawn from the distributions described in Table 1 using the Sobol′ sequence. The input parameters are T: total population, $L$: population distribution, S: impact function threshold shift, and $H$: hazard events bootstrapping.

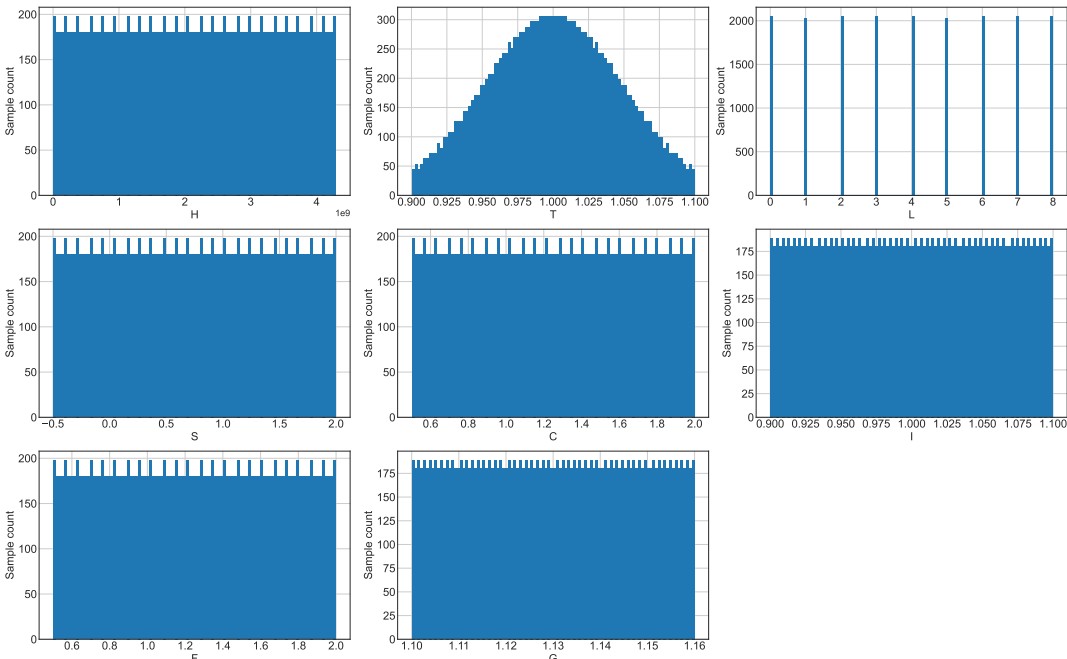

**Figure A4.** Samples for the uncertainty analysis of the adaptation options appraisal in Sect. 3.2.2 for the input parameters drawn from the distributions described in Table 1 using the Sobol′ sequence. The input parameters are $H$: hazard events bootstrapping, T: total population, $L$: population distribution, S: impact function intensity threshold shift, C: cost of adaptation options, I: hazard intensity change, F: hazard frequency change, and G: population growth.

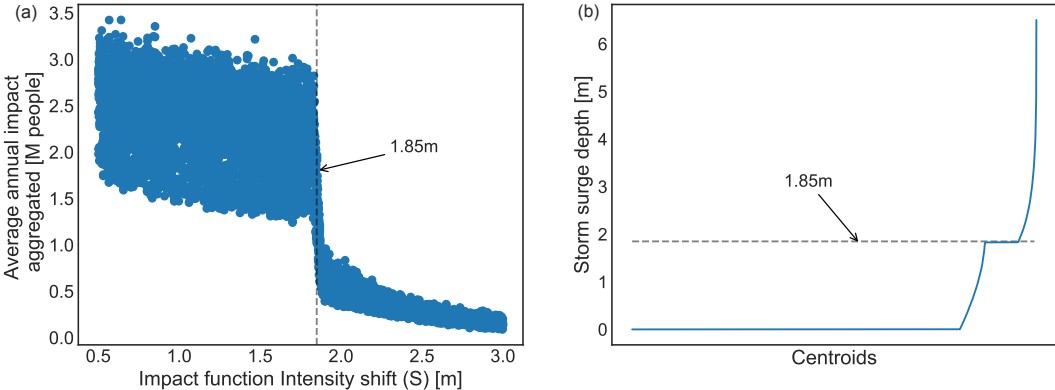

**Figure A5.** (a) Annual average impact averaged over all exposure points in millions (M) affected people as a function of the impact function threshold shift uncertainty (S) in meters, and (b) storm surge intensity in meters (m) of all events at each location (centroid) from the original case study. A non-linear change in intensity at $\sim 1.85$m is indicated by a dashed line.

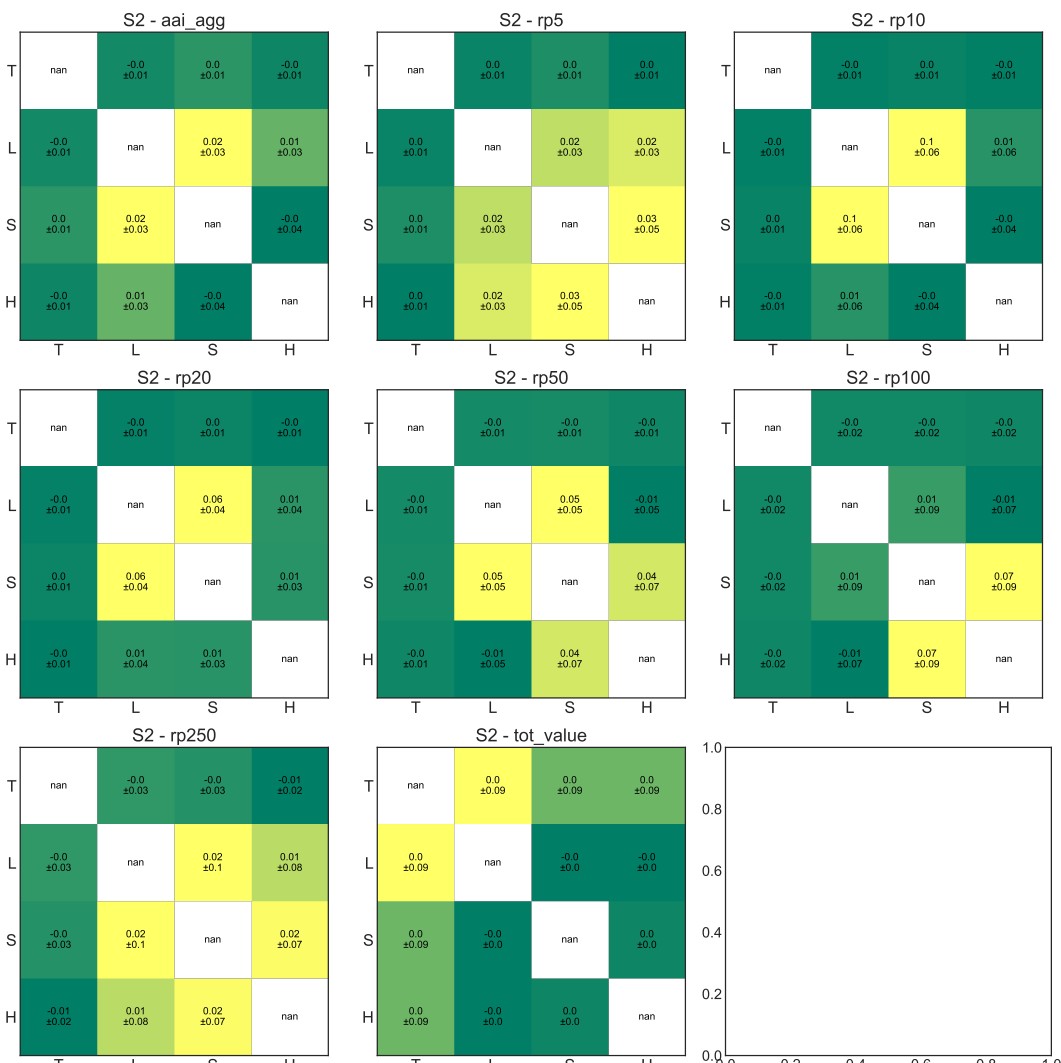

**Figure A6.** Second-order Sobol′ sensitivity indices (S2) for different storm surge risk metrics: average annual impact aggregated over all exposure points (aai_agg), impact for return periods (rp) $5, 10, 20, 50, 100, 250$ years and the total exposure value (tot_value). The input parameters (c.f., Table 1) are T: total population, L: population distribution, S: impact function intensity threshold shift, and H: hazard events bootstrapping.

*Author contributions.* Conceptualization: C.M.K. and A.C., S.M., D.N.B., E.S., L.O., J.W.M. ; Writing original draft: C.M.K. and A.C. ; Writing review and editing: C.M.K. and A.C., S.M., D.N.B., E.S., L.O., J.W.M., A.R. ; Data curation: C.M.K. and A.R. ; Formal Analysis: C.M.K. ; Software: C.M.K. and E.S. ; Resources: D.N.B. ; Visualization: C.M.K. and A.C. ; Project Administration: C.M.K.

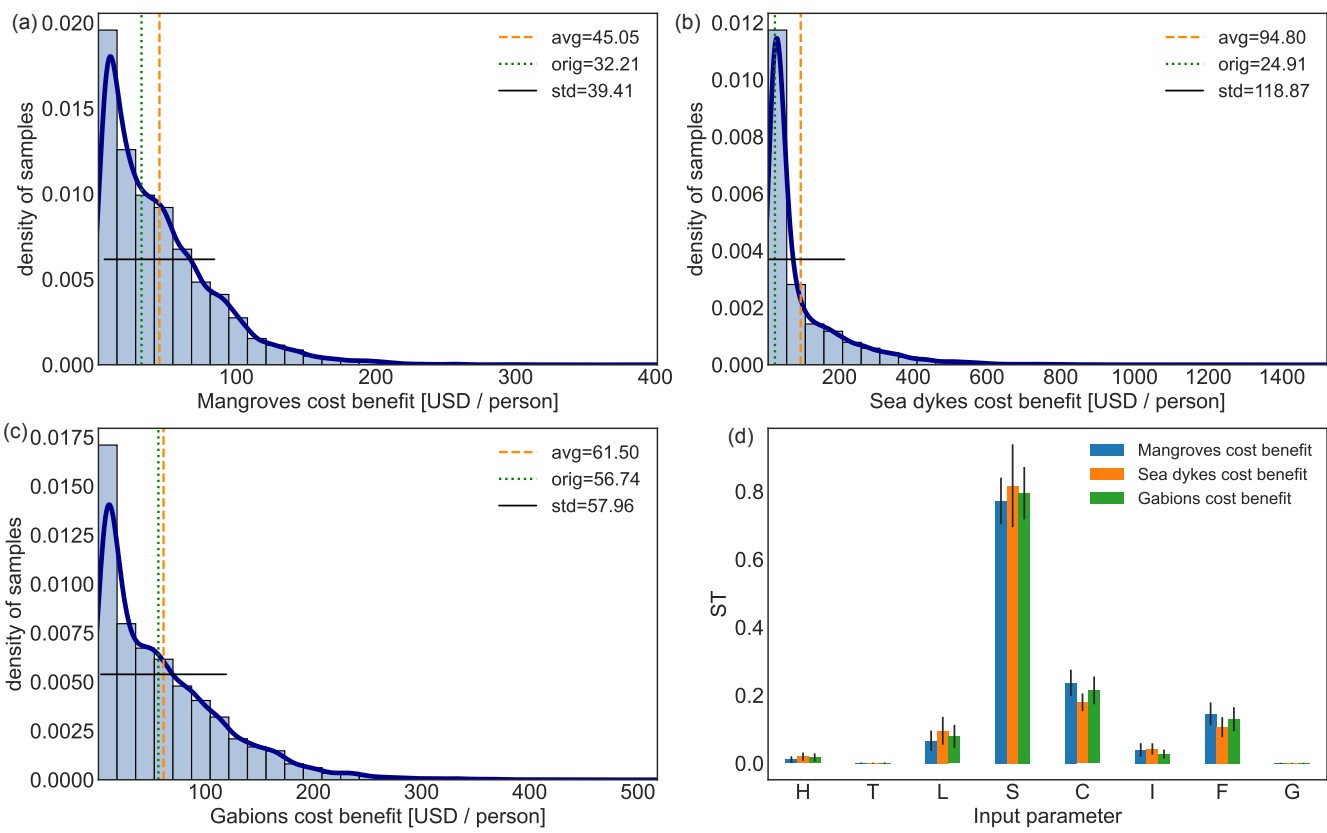

**Figure A7.** Uncertainty distribution (histogram bars) for the ratio of cost to benefit of the three adaptation options (a) mangroves, (b) sea dykes, and (c) gabions. In addition, (d) the total order Sobol' sensitivity indices (ST) for the three adaptation options. Cost to benefit ratios panels include the original case study value (dotted green vertical line), average (dashed orange vertical line), standard deviation (solid black horizontal line), and kernel density estimation to guide the eye (solid dark blue line). The total order Sobol' sensitivities are shown with a black bar indicating the 95th percentile confidence interval. The input parameters (c.f., Table 1) are $H$: hazard events bootstrapping, T: total population, $L$: population distribution, S: impact function intensity threshold shift, C: cost of adaptation options, I: hazard intensity change, F: hazard frequency change, and G: population growth.

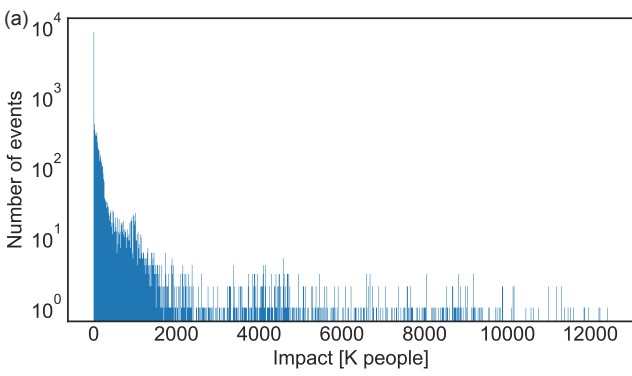 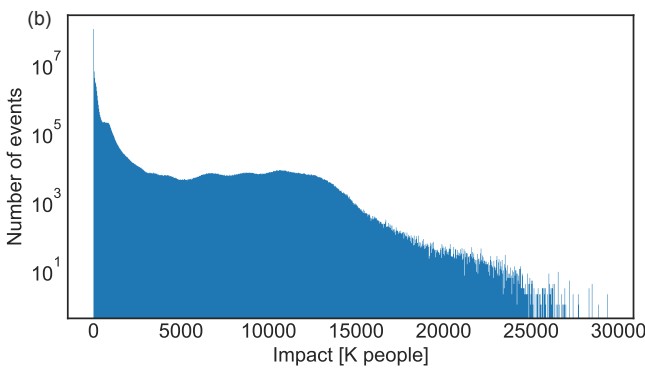

**Figure A8.** Histogram of the number of storm surge events in the probabilistic set by their impact (in thousands (K) of affected people) in Vietnam for present climate conditions (2020) for (a) the original case study probabilistic set, and (b) union of the probabilistic sets for all samples of input parameters considered in Sect. 3.2.4 (c.f. Fig. A3). Note the logarithmic scale of the vertical axes.

*Competing interests.*  The author declare no competing interests.

*Acknowledgements.*  The authors are grateful to Moustapha Maliki and Evelyn Mühlhofer for valuable discussion at the start of this project. This project has received funding from the European Union's Horizon 2020 research and innovation programme under grant agreement No 821010 and under grant agreement No 820712.

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
