# Peer review of "Uncertainty and sensitivity analysis for probabilistic weather and climate risk modelling: an implementation in CLIMADA v.3.1.0"

_Geoscientific Model Development, 2021_

## Author Response (AR1)

>> The manuscript presents an interesting advancement to the CLIMADA risk modelling platform to enable rigorous assessment of uncertainty propagation through risk models. This new functionality will be of interest to a broad range of CLIMADA users as well as serve as inspiration for developers of other modelling platforms to implement similar advancements. I think the manuscript is overall well structured and well written. The discussion and outlook section is in my opinion particularly thoughtful and provides a very good account of benefits and limitations of current approaches to uncertainty quantification. I think this manuscript and the underpinning work offers a practical contribution to accelerate the uptake of good practices in the risk modelling community, as well as a contribution to foster further discussion on the need of better handling of model uncertainties. I would thus recommend the manuscript for publication after minor revisions. Below are some suggestions for improvement which I hope the authors may consider in preparing a final version.

-- Thank you very much for the thorough review and the positive feedback! Below we reply directly to the valuable and insightful comments.

>> Referee comments
-- Authors replies
Blue: text additions; Red: text removal
* * *
>> MAIN POINTS

>> [1] sampling of hazard component
I understand that the hazard component in the CLIMADA platform is based on directly loading a hazard event set (p 4 line 85). I suppose CLIMADA users will use different approaches to generate the event set, sometimes relying on data, other times relying on model simulations. For example, in the case of flood risk assessment, people typically use dynamic hydrological-hydraulic models to produce maps of river flow peaks or flood inundation depths over the modelled domain. In such cases, I understand that the current implementation of CLIMADA does not allow to explicitly include the input factors of the dynamic hazard model into the uncertainty and sensitivity analysis (UA/SA).

In other words, users cannot sample the input factors of the hazard models and propagate uncertainties, instead they will have to define ways to perturb the hazard model outputs directly.

-- Thank you for this comment. We realize that our wording may be misleading and can create a misunderstanding. The function for sampling an input variable (hazard, exposure, impact function, measure) does, in fact, lend itself to wider applications than solely the CLIMADA core code and its functionality. A user could for instance write a wrapper around a bespoke dynamical hazard model which takes as input the uncertain input parameters, and

produces as outputs the resulting hazard object. This wrapper can then be used in the CLIMADA uncertainty and sensitivity analysis module. We note that this might then lead to prohibitive computation times in case the dynamical model was costly. A way around would be to pre-compute a set of dynamical model hazard outputs for a fixed set of input factors, and then use in the CLIMADA unsequa module only those precomputed outputs. This would trade CPU computation time for memory. This was done, for instance, in the example for the LitPop module to obtain a pre-computed set of exposures with spatially distributed perturbations. A detailed paragraph and an example were added to clarify this point.

130     measure ($M$) in a container input variable called entity ($T$). Note that further input variables might be added in future versions of CLIMADA. Each of these input variables comes with any number of uncertainty input parameters $\alpha$, distributed according to an independent probability distribution $p_\alpha$.  An input variable can have any number of uncertainty input parameters, and there is no restriction on the type of probability distributions (e.g. uniform, Gaussian, skewed, heavy-tailed, discrete, etc.). In the current implementation any distribution from

135     the *Scipy.stats* Python module (Virtanen et al., 2020) is accepted. The input parameters can define any variation or perturbation of the input variables (e.g., initial conditions, boundary conditions, forcing inputs, resolutions, normative choices, etc.).[2] Note that the choice of the variation and the associated range and distribution can substantially affect the results of an uncertainty and sensitivity analysis (Paleari and Confalonieri, 2016). Ideally this modelling choice should be made based on solid background knowledge. However, the latter is often lacking or highly uncertain; in such cases, we encourage users to explore how the

140     results may vary with alternate distributions and choices of input parameters. It is thus not always about deriving definitive quantitative values describing the deviation of the climate risk model's output from the "real" value, but also about assessing the robustness, sensitivity and plausibility of the model output under clearly defined assumptions.

 Overall, the user must define one method for each of the uncertain input variables $X$, which returns the input variable's value $X(\alpha_1, \alpha_2, \ldots)$ for each valid value of the associated uncertain input parameters $\alpha_1, \alpha_2, \ldots$. The latter are univariate

145     random variables distributed according to $p_{\alpha_1}, p_{\alpha_2}, \ldots$. In order to support the user, a series of helper method are implemented in the unsequa module (c.f. Appendix B). This general problem formulation allows for generic uncertainty parametrization, with broadly speaking two types of approaches: an input variable is directly perturbed with statistical methods, or the underlying

model used to generate the input variable is fed with the uncertain parameters. Note that each input variable is independent, and thus either approach can be used for different input variables for the same study (c.f. the illustration case study in Section

150     3.1)

160      As another example, we are interested in the risk of floodplain flooding for gridded physical assets in the Congo basin. The flood hazard is generated from a floodplain modeling information system (FMIS) with uncertainty parameters describing the uncertainty in the geospatial data, the temporal data, the model parameters (Mannings) and the hydraulic structure, such as shown in Merwade et al. (2008). These input parameters are used directly as uncertainty input parameters for the unsequa model, with a wrapper method returning a CLIMADA hazard object produced from the FMIS flood inundation

165     map. In addition, the exposures are obtained by interpolating and down-scaling satellite images to a resolution $r$ (Eberenz et al., 2020). The sensitivity and robustness to the resolution choice is modelled by pre-computing exposures at resolutions $r = 50as, 100as, 150as$. The uncertainty parameter is then $r$, with a uniform choice distribution between the pre-computed values. Finally, we consider all assets to be described by a single impact function, which is derived from three different case studies found in literature. The impact function's uncertainty is defined as a uniform choice distributed parameter $u \in [1, 2, 3]$ corresponding to the selection

170     of one of the three impact functions.

>> I think in many cases this may be quite difficult as it may not be obvious at all how one can perturb spatially-distributed outputs (for example flood depth maps) in a way that is physically consistent and respect plausible spatial patterns. The authors only touch upon

this on p. 6 line 133 where they say: "this modelling choice should be made based on solid background knowledge". Honesty I find this statement a bit simplistic - reality is very often we do not have that knowledge and defining a reasonable way to sample spatially-distributed variables can be the most critical and time-consuming aspect of setting up the UA/SA. I am not suggesting the authors should solve this issue (far from it!) but I think they should point out this is a very important and critical step, and a big area for future research and development. In my opinion this question - how do we sample complex and spatially-distributed variables in a meaningful way? - is one of the key research questions that the UA/SA community will need to work on if we want to move on to the next stage of applying this type of techniques to complex models.

-- Thank you for these very pertinent comments. We agree that this is certainly a very central point for future research. We added a few remarks on this point.

> While we showed that quantitative uncertainty and sensitivity is a significant step to improve the information value of climate risk models, we stress that not all uncertainties can be described with the shown method (see e.g., Appendix D for a discussion on event uncertainty). Indeed, only the uncertainty of those input parameters that are varied can be quantified, and even for these input parameters, defining the probability distribution is subject to strong uncertainties, often being based only on educated guesses. Yet, the choice of probability distribution can have a strong impact on the resulting model output distribution and sensitivity (Pianosi et al., 2016; Saltelli et al., 2019; Otth et al., 2022)(Paleari and Confalonieri, 2016; ?; Otth et al., 2022). In addition, it is often not evident how to perturb the input variables, since one does not always have access to the underlying generating model, and it is otherwise difficult to define physically consistent statistical perturbations of geospatial data. Moreover, there is a large part of climate risk models uncertainty that is not even in principle quantifiable (Beven et al., 2018b; Knüsel, 2020a)(Beven et al., 2018b; Knüsel et al., 2020). When building a climate risk model, a number of things must be specified, such as the model type, the algorithmic structure, the input data, the resolution, the calibration and validation data etc. These choices

> Defining the appropriate input variable uncertainty definition, and identifying the relevant input parameters for a given case study is not a trivial task. In general, only a small subset of all possible parameters can be investigated Dottori et al. (2013); Pianosi et al. . In order to identify the relevant parameters and defining the input variables' uncertainty accordingly, one can for instance use an assumption map (Knüsel et al., 2020), as presented for CLIMADA in Otth et al. (2022). Otth et al. (2022). Another general strategy is to proceed iteratively: a first broad sensitivity analysis is used to identify the most likely important uncertainties, followed by a more detailed uncertainty and sensitivity analysis for full quantification.

>> Related to this point, I also wonder how difficult it would be to enable users to load into CLIMADA an ensemble of perturbed event sets, in place of applying perturbations within the platform? If this was viable, users could do the sampling and Monte-Carlo simulations of the hazard model outside CLIMADA, and then use the platform to combine the hazard samples with exposure and vulnerability samples and perform the final calculations of sensitivity indices.

-- Thank you for this very practical question. As discussed above, this is currently already possible. With the unsequa module, the perturbation model of the input variables exposures, hazard, impact function and measures is very free. For instance, in this manuscript we show that we can sample the spatial distribution of exposures outside of CLIMADA, and then use a discrete uncertainty parameter to select the different samples in the uncertainty and sensitivity analysis. Similar can be done for example with a hydraulic model with continuous uncertainty distributions of e.g., the manning parameter. Then, the user might for instance first build the global sample for all input parameters (including those for the exposure and impact function), and then run the hydraulic model for all the different sampled values of the manning parameter to obtain a hazard set to use in the uncertainty and sensitivity analysis. In order to support the users in quickly and easily defining a broad

variety of uncertainty input variables for the unsequa module we added an appendix describing a set of helper methods.

**Appendix B: Helper methods**

[revised manuscript text omitted]

>> p. 7 line 159: "In general, it is recommended to use..." I disagree. GSA literature shows that the sample size needed to achieve reasonable approximation of sensitivity indices significantly vary with GSA method but also, for the same method, with the case under study (see for example Sarrazin et al. 2016). For variance-based methods, "adequate" sample size can range from 1,000 up to 10,000 (or more) times the number of input factors (see for example Figure 5 in Pianosi et al. (2016)) so I would really avoid giving readers a "one-fit-for-all" recommendation, which may be misleading.

As pointed out in Sarrazin et al. 2016, a better approach is to start with the sample size that one can afford to generate reasonably efficiently, and then check the robustness of the estimated sensitivity indices. If the key conclusions about the input ranking or screening are unambiguous despite the uncertainty in estimated indices, fine, otherwise one should either generate more samples of fall back to using a more frugal GSA method. This is indeed what the authors themselves do when they calculate confidence intervals and check that they do not overlap significantly. So I would suggest to revise this paragraph bringing in this discussion and potentially anticipating or referring to the later description of how confidence intervals can be used to estimate robustness of the results.

-- Many thanks for this remark. We did not imply to give a "one-fit-for-all", but rather an indication of what we found is often sufficient in the context of CLIMADA. We now understand that this might be misleading. We removed the number '1000' suggestion and added a brief discussion on choosing the number of samples.

> 2.3.1, one would create $N$ global samples $x_n = (t_n, s_n, a_n)$ with $n \in [1, \ldots, N]$. One sample thus corresponds to a set of three numbers in this case.  Choosing the correct number of samples is a notoriously difficult task
> 190 (Iooss and Lemaître, 2015; Sarrazin et al., 2016). One generic approach is to start with a sample size that one can afford to generate reasonably efficiently (e.g., $N \sim 100D$  ), and then check the confidence intervals of the estimated sensitivity indices (c.f. Section 2.3.5). If relative values of the estimated indices are too ambiguous due to the overlap of confidence intervals to draw key conclusions, one should either generate more samples, or use a more frugal method (e.g., reduce the number of input parameters
> 195 ) (Sarrazin et al., 2016).
* * *
>> p. 7 line 182 and Section 2.3.5 - following up on my previous comment: I would also suggest to add some more information about the confidence intervals, specifically: (1) how are they derived? no need to go into the details but at least say in one sentence what's the key idea/methodology (bootstrapping?) to derive them. (2) how can they be used? again I would briefly explain to the reader how confidence intervals should be interpreted and used (I am thinking something like the discussion of Figure 4 in Noacco et al 2019 or even shorter).
I insist on this point as in my experience the choice of the sample size is one of the most confusing for GSA users, especially when doing GSA for the first time, so I think it is important to give sound and clear advice on this!

-- Thank you for pointing out this lack of precision from our side. Following the suggestion, we added a brief note on this point.

> We remark that no direct evaluation of the convergence of this quasi Monte-Carlo scheme is provided in the unsequa module, as it is not generally available for all the possible sampling algorithms available through the *SALib* package. Instead, the sensitivity analysis algorithms, to be described in Sect. 2.3.5 below, provide confidence intervals. In SALib, confidence intervals relate to the bounds which cover 95% of the possible sensitivity index value, estimated through bootstrap resampling. These
> 220 can be used as a proxy to assess the convergence of the uncertainty analysis. If the intervals are large and overlapping, the result is likely not robust and the number of samples should be increased.
* * *
OTHER MINOR POINTS

>> p. 2 line 31: "In practice, the quantification of risk with climate risk model..." A recent paper that also makes this point and shows how GSA can be used for the evaluation of impacts models, particularly when fit to historical observations may not be a sufficient criterion, is Wagener et al (2022)
-- Thank you for the very pertinent references. We added it to the relevant argument.
* * *
>> p. 2 line 48: "an analytical treatment is often not possible", I agree though when it is possible it should be considered as the primary route to SA. I think a useful reference here may be Norton (2015) which covers some of the analytical approaches to SA
-- We thank you for this valuable comment. We amended the text to make this clearer.

> Among the established methods proposed by the scientific literature to quantitatively treat uncertainties in model simula-
> 50 tion are uncertainty and sensitivity analysis (Saltelli, 2008).  While for both methods an analytical treatment is preferable (Norton, 2015), it is often not possible. Therefore, numerical Monte-Carlo or Quasi-Monte-Carlo schemes
* * *
>> p. 2 line 50: "uncertain input parameters" Here and everywhere else, the authors use the term "parameters" to refer to the uncertain inputs that are varied in the uncertainty and sensitivity analysis. I find this terminology potentially misleading, as some may interpret the term "parameters" in a narrower and more specific sense (for instance in dynamic systems terminology people tend to distinguish the model "inputs" into initial conditions, boundary conditions, forcing inputs, parameters - hence using "parameters" in a very specific sense). Indeed on page 4 (L. 108) the authors mention more generic "input variables and parameters definition". I would suggest either clarifying what the term "parameters" refer to or, as commonly done in the GSA literature, use the term "input factors" instead.

-- We indeed had difficulties finding a terminology that will be understandable by everyone in this interdisciplinary field.  We add clarifications at multiple points in the text in order to make our use of the term clear. We had settled on the terminology "uncertainty parameter" to define any univariate random variable that is sampled from, and "input variable" to define the inputs of the CLIMADA model (exposures, hazard, impact function, measures). We felt that "input factors" would be potentially confusing, as it is not clear whether CLIMADA variables or random variables are meant. Thus, we follow the "parameter" terminology of for instance Merwade et al. (2008).

135 distributions (e.g. uniform, Gaussian, skewed, heavy-tailed, discrete, etc.). In the current implementation any distribution from the *Scipy.stats* Python module (Virtanen et al., 2020) is accepted. The input parameters can define any variation or perturbation of the input variables (e.g., initial conditions, boundary conditions, forcing inputs, resolutions, normative choices, etc.). [2] Note that the choice of the variation and the associated range and distribution can substantially affect the results of an uncertainty and sensitivity analysis (Paleari and Confalonieri, 2016). Ideally this modelling choice should be made based on solid background

The general workflow of the new uncertainty and sensitivity quantification module unsequa, illustrated in Fig. 1, follows a Monte-Carlo logic (Hammersley, 1960) and implements similar steps as generic uncertainty and sensitivity analysis schemes

110 (Pianosi et al., 2016; Saltelli et al., 2019). It consists of the following steps:

  – *Input variables and input parameters definition.* The probability distributions of the uncertain input parameters (random variables) are defined. They characterize the input variables  hazard, exposure and impact function for risk assessment, and additionally adaptation measure for adaptation options appraisal – of the climate risk model CLIMADA.
* * *
[2] In literature, the terminology "input factor" instead of "input parameter" is also used. Here we shall use exclusively the terminology "input parameter" for numerical random variables for which a random sample can be drawn, and "input variable" for the inputs to the CLIMADA model.
* * *
>> p. 6 line 133: "Note that the choice of the variation range ..." A reference with some good examples of this problem that could be added here is Paleari and Confalonieri (2016).

-- Thank you for point us towards this interesting reference which we added to the manuscript.
* * *
>> p. 8 line 191: "long single impact computation time". What are the factors that control computation time in CLIMADA? I understand that the platform does not execute any dynamic hazard models but instead it directly loads a hazard event set. Hence, even if the event set was generated with repeated executions of a (expensive) hazard model, the complexity of these calculations should not affect CLIMADA computing time. Am I right? If so, then I suppose CLIMADA computing time should be mainly controlled by the chosen spatial resolution - again, is this correct? I think these points would be worth clarifying.

-- You are indeed right. If the model is setup correctly, in most cases the main driver of computation times is the model's spatial resolutions, the size of the event set, and the number of adaptation measures. However, if the model is coupled to another dynamical model (e.g. a flood model), the computation time could also be driven by this external model. We added a remark on this in the manuscript.

225     periments showed that the computation time scales approximately linearly with the number of samples $N$, and is proportional

to the time for a single impact computation. The  latter is mostly defined by the size of the exposure (i.e.,

depends on the resolution, size of the considered geographical area, ...) and the size of the hazard (i.e., depends on the number

of events, the centroid's resolution, etc.). In case the input variables are generated using an external model (e.g., a hydrological

flood model for the hazard), the computation time is also proportional to the external model run time. For complex models, this

230     can be prohibitively long. In such case, one can pre-compute the samples for the given input variable, thus trading CPU time

for memory (c.f. Litpop example in Section 3.2.1, and the helper methods in Appendix B). The number of samples $N$ in turn

scales with the dimension $D$ (i.e., the number of input parameters) depending on the chosen sampling method. For the default
* * *
>> p. 8 line 209: "Note that it is perfectly valid to use different sampling..." I am a bit confused by this remark. Yes it is possible to use different sampling strategies but why would one want to do that? My take on this is that the direction we are moving towards is quite the opposite, that is, the GSA community is increasingly focusing on UA/SA methods that work on the same generic input-output sample, so that we can reduce the effort of generating the sample (which is often the computational bottleneck of this type of analysis) and make the most of it for both uncertainty quantification and attribution. See for example discussion of "given-data approaches" in the introduction of Borgonovo et al (2017).

-- Thank you for pointing out this misleading comment of ours. We just wanted to point out that the presented module does not prescribe any fixed choice. We added a sentence to make it clear that this is a comment of technical nature, and not a methods recommendation.

save-guard checks have been implemented in the unsequa module. Note that it is  technically valid to use different

255     sampling algorithms for the uncertainty, and for the sensitivity analysis. For example, one can first use sampling algorithm A

to perform an uncertainty analysis, i.e., steps from Sects. 2.3.1 - 2.3.4. Then, use another sampling algorithm B as required for

the chosen sensitivity index algorithm to perform the sensitivity analysis, i.e., steps from Sects. 2.3.1-2.3.3 and 2.3.5, 2.3.6.

However, in practice, since generating samples is often the computational-time bottleneck, it is more convenient to use the

same methods so that the same samples can used for both steps (Borgonovo et al., 2017).
* * *
>> p. 11 line 296: "the distribution of uncertainty... is bi-modal" In terms of GSA, this is interesting as the use of variance-based indices with output distributions that are multi-modal or highly skewed may be a bit critical. In fact the underpinning assumption of variance-based sensitivity analysis is that variance is a good proxy of output uncertainty (in other words, variance is a good statistic to synthetically characterise the shape of the output distribution). This assumption is perfectly fine for symmetric distributions whereas it becomes more and more questionable with multi-modal or highly skewed ones. In such case a different GSA method may be more appropriate (see for example discussion in Pianosi and Wagener 2015). I am not suggesting that the authors perform any further analysis but maybe they may consider making a comment here or think about this in future developments.

-- Thank you for this remark, as this is one of the main points we would like to convey with our example calculation. Blindly looking at mean value and standard deviations, or computing variance-based sensitivity indices is not enough for a thorough understanding of the model. One really must look at the full uncertainty distribution. We added a comment in the discussion that for the illustrative case in this paper, it would have been beneficial to continue the study by using non-variance-based methods.

 Ideally, we should choose the sensitivity method best suited for the data at hand. In our case, the uncertainty distribution is strongly asymmetric (c.f., Fig. 2), thus a density-based approach would be best (Pianosi and Wagener, 2015; Borgonovo, 2007; Plischke et al., 2013). However, this would require generating a

360     new set of samples, and for the purpose of this demonstration we used the unsequa default variance-based Sobol method. Note that despite the questionable use of variances to characterize sensitivity for multi-modal uncertainty distributions, the derived indices prove useful to better understand the results from the case study at hand.

We thus computed the total-order and the second-order Sobol$'$ indices (Sobol$'$, 2001) for all the input parameters $T, L, H, S$.
* * *
>> p. 13 line 314: "strong correlations" should be "strong interactions"
-- This is an excellent suggestion and will be implemented as such.

>> In their manuscript "Uncertainty and sensitivity analysis for probabilistic weather and climate risk modeling: an implementation in CLIMADA v.3.1.0", Kropf et al present a new module to the climate risk modeling platform CLIMADA. This module is designed specifically to calculate global-scale uncertainty and sensitivity analyses related to various natural hazards and impacts. I can foresee that this new functionality will be of interest to a broad range of CLIMADA and catastrophe model users, and that this new feature will be on the forefront of (academic) risk modeling for the next years to come. I therefore recommend publication of this article after some minor comments have been addressed, see below.

-- Thank you very much for the excellent review and the positive feedback! Below we reply directly to the improvement suggestions and general comments.

>> Referee comments
-- Authors replies
Blue: text addition
Red: text removal
* * *
>> Main comments

>> My main comments are regarding the hazard set used in this analysis. While I understand that this is (probably) the exact same hazard dataset as was used in Rana et al (2021), I still think it's good to provide a bit more information on the construction of this hazard set. Particularly since this manuscript focuses on uncertainty analysis.
-- Thank you for pointing out the lack of clarity concerning the hazard element of the case study. We implemented all the proposed changes as listed below to improve the understanding of our manuscript.

>> (Line 270). Could you please explain in more detail what the event set is; what input data is the set based on, how exactly is this event set perturbed in CLIMADA?
To my understanding, CLIMADA will only perturb tropical cyclone tracks present in this event set. Does this mean that unprecedented events won't be simulated by CLIMADA? How certain are you that the resulting set of tropical cyclone events presents the full range of all possible events near Vietnam?
-- The probabilistic event set is generated using a random perturbation algorithm of the historical tracks from the IBTrACS dataset. The windfields are then generated with the algorithm from Holland et al. 2008. The event is perturbed by bootstrapping for the uncertainty assessment. Thus, unprecedented events are simulated in this event set. We cannot be certain that the resulting set of tropical cyclones events presents the full range of all possible events near Vietnam, and to our knowledge no work exists that could claim so. This is precisely why there is a need for uncertainty and sensitivity. We added a note on this point in the manuscript.

For the hazard, we apply a bootstrapping technique, i.e., uniform re-sampling of the event set with replacement, in order to account for sample estimates uncertainties. Since the default Sobol' global sampling algorithm requires repeated application of the same value of any given input parameter, here we define $H$ as the parameter that labels a configuration of the re-sampled events. Errors from the hazard modelling (c.f. Appendix C) are not

320    further considered here. A more detailed study might want to explore further uncertainty sources, such as the windfield model, the hazard resolution or the random set generation algorithm.

We would also like to clarify two points that might be unclear. CLIMADA is a risk framework model. Thus, it is not bound to one hazard model. The sentence "CLIMADA will only perturb tropical cyclone tracks present in this event set.» is thus correct in the narrow sense that in the presented case study, we used perturbed tropical cyclone track set. It is not correct in the general sense though, as any other tropical cyclone model can be used in CLIMADA. See eg. Meiler et al. (2022) https://doi.org/10.21203/rs.3.rs-1429968/v1 .

Furthermore, the bootstrapping perturbation is a choice made here mainly to illustrate the module. It allows to capture the sampling uncertainty within the probabilistic event set. Other forms of uncertainty could be explored if desired. For instance, one could generate different probabilistic sets with different parameters for the random perturbation algorithm. Or one could use another probabilistic event set, such as the one presented by Bloemendaal et al. (2021).

We understand that these points were not clearly stated enough in the manuscript, as noted also by the other referee, Francesca Pianosi. In response, we added substantial clarifications (for the detailed text changes, please see the response to the comment by Francesca Pianosi).

>> Section 3.3.1. Could you please elaborate a bit on how exactly the original case study uses the parameters from Knutson et al. (2020)? Does the future-climate event set also contain information on shifts in tracks/genesis locations? And are the changes in intensity uniformly applied across the track, or does this only apply to the peak intensity?

-- The parameters are only used to homogeneously scale the tracks intensity and frequency by basin. There is no track location change included. This is a limitation of the study by Rana et al. 2022, which was partially explored in the current uncertainty analysis by the bootstrapping uncertainty. We added a note on this in the manuscript.

et al. (2021). Future climate hazard sets were created for two Relative Concentration Pathways (RCP) (Pachauri et al., 2015), RCP6.0 and RCP8.5, based on parametric estimates. For each storm, the intensity and frequency where homogeneously shifted by a multiplicative constant derived from (Knutson et al., 2015) based on the storm's Safir-Simpson

580    category.

>> I like the final sentence of Section 3.2.6 "Together, these results hint to potentially hidden high-impact events in unexpected areas" (line 338), but it also feels like a cliffhanger! What events are we talking about, could you please give an example of such event in the text/figure?
-- We are glad to have captured your interested here! We added an example as suggested.

395 distribution $L$. Furthermore, while for shorter return periods, the largest total-order sensitivity index is the impact function threshold shift $\text{ST}_S$, for longer return periods the sensitivity to the population distribution $\text{ST}_L$ gets larger as shown in Fig. 3 (c). This might be because stronger events with large return periods consistently have larger intensities than the maximum threshold shift of $3m$. Together, these results hint to potentially hidden high impact events in unexpected areas  (e.g., a large storm surge in the less densely populated southern tip of Vietnam could affect a large number of people).

>> Line 325: For me, it's unclear why this number is 1.85m. Does this have to do with protection standards?
-- Thank you for the interesting suggestion. Actually, we were not able to clarify the origin of this number within this project. We added a brief discussion on it in the manuscript.

impact is discontinuous. Thus, the bi-modality of the uncertainty distributions, while caused by uncertainty in the impact function roots in the modeling of the storm surge hazard footprints.  Further research beyond the scope of this paper would be need to understand whether this value of 1.85m has a physical

385 origin (e.g., landscape features or protection standards), or is due to a modelling artefact. However, despite the discontinuity, the patterns are as expected: an impact function with a step at $0.5m$ results in many more people being classified as affected than when the steps is at 3m (in the latter case, only particularly large storm surges would results in people being affected). For planning purposes, the lower end of this impact function shift is most relevant – even 0.5m depth of storm surge

390 can be dangerous for people - so the higher mode of the distribution in Fig. 2 is most relevant.

>> I recommend to acknowledge somewhere that the results obtained here are solely for storm surge, and that including wind and precipitation can alter the outcomes.
-- Thank you for the suggestion. We added a note in the conclusions.

460 the results of the study should have been subject to considerable uncertainty. The need of uncertainty and sensitivity analysis was identified within the original study, but deemed out of scope. This was in part due to the absence of a comprehensive and easily applicable scheme, now resolved with the uncertainty and sensitivity quantification module presented here. In addition, a full-fledged uncertainty and sensitivity analysis leads to a large amount of additional data to process. Indeed, the results shown in this section considered only a small subset of the original case study, which, among others, also considered the impact of

465 tropical cyclone wind gusts, and the impact of wind and surge on physical assets in dollars. Nevertheless, the benefits of an uncertainty and sensitivity analysis are manifest. On the one hand, it provides a much more comprehensive picture on risk from storm surges and the benefits of identified adaptation measures. On the other hand, it allows to identify the main shortcomings

>> (Very) minor comments

>> Please check the reference style in lines 53 and 436, and the reference in line 347.
-- Thank you for the very attentive reading. We updated the styles appropriately.

>> Please consider writing "exposure" throughout the manuscript rather than "exposures". To my understanding, exposure is the more commonly used term to address the full set of exposed elements, and the use of exposures leads to some grammatically incorrect sentences in the manuscript(e.g. line 26)
-- Thank you for this remark. In turns out that the class name in the CLIMADA code-base is "Exposures" and we thus used a similar writing. However, we agree that "exposure" is less confusing, thus we adapted the text accordingly.

>> Line 301 - 303 is very hard to follow. Please consider breaking this sentence up in two or rewriting this sentence.

-- We understand that the sentence is too convoluted and rewrote the argument as two sentences.

350     The bi-modal form of the impact uncertainty distribution is interesting, as one could rather expect statistical white or colored noise (e.g., Gaussian or power-law  distributions). As a proof-of-consistency  that this is not due to a computational setup error, we verified that the distribution of the total asset value, shown in Fig. 2 (d), aligns with the parametrization of the  exposure uncertainty (c.f. Table 1). For a better understanding of the obtained uncertainty distributions, and in particular understand the bi-modality, let us continue with the sensitivity analysis.

>> "Adaptation" is misspelled as "Adpatation" in multiple instances.
-- Thanks! We corrected the wrong spelling in the manuscript.

---

## Author Response (AR2)

Dear Chris Folberth,

thank you for the attentive care to the open data policy, this is highly appreciated. We realized that the github repository containing the scripts to reproduce the data from the case study Rana et al. (2021) had wrong permission settings and was thus not public. This has been solved. In addition, we uploaded the hazard data to the ETH research collection. The text for the data and code availability has been updated accordingly.

We hope the manuscript now fulfills all requirements.

Kind regards,
Chahan Kropf and co-authors.

---

## Author Response (AR3)

Dear Chris Folberth,

thank you for the attentive care to the open data policy, this is highly appreciated. All the scripts have now been deposited in a frozen version.

We hope the manuscript now fulfills all requirements.

Kind regards,
Chahan Kropf and co-authors.